# Is forgetting less a good inductive bias for forward transfer?

**Jiefeng Chen** *
Department of Computer Science
University of Wisconsin-Madison
Madison, WI 53706, USA
jiefeng@cs.wisc.edu

**Timothy Nguyen & Dilan Gorur & Arslan Chaudhry** †
DeepMind
Mountain View, CA
{timothycnguyen,dilang,arslanch}@deepmind.com

## Abstract

One of the main motivations of studying continual learning is that the problem setting allows a model to accrue knowledge from past tasks to learn new tasks more efficiently. However, recent studies suggest that the key metric that continual learning algorithms optimize, reduction in catastrophic forgetting, does not correlate well with the forward transfer of knowledge. We believe that the conclusion previous works reached is due to the way they measure forward transfer. We argue that the measure of forward transfer to a task should not be affected by the restrictions placed on the continual learner in order to preserve knowledge of previous tasks. Instead, forward transfer should be measured by how easy it is to learn a new task given a set of representations produced by continual learning on previous tasks. Under this notion of forward transfer, we evaluate different continual learning algorithms on a variety of image classification benchmarks. Our results indicate that less forgetful representations lead to a better forward transfer suggesting a strong correlation between retaining past information and learning efficiency on new tasks. Further, we found less forgetful representations to be more diverse and discriminative compared to their forgetful counterparts.

## 1 Introduction

Continual learning aims to improve learned representations over time without having to train from scratch as more data or tasks become available. This objective is especially relevant in the context of large scale models trained on massive scale data, where training from scratch is prohibitively costly. However, the standard stochastic gradient descent (SGD) training, relying on the IID assumption of data, results in a severely degraded performance on old tasks when the model is continually updated on new tasks. This phenomenon is referred to as catastrophic forgetting (McCloskey & Cohen, 1989; Goodfellow et al., 2016) and has been an active area of research (Kirkpatrick et al., 2016; Lopez-Paz & Ranzato, 2017; Mallya & Lazebnik, 2018). Intuitively, the reduction in catastrophic forgetting allows the learner to accrue knowledge from the past, and use it to learn new tasks more efficiently – either using less training data, less compute, better final performance or any combination thereof. This phenomenon of efficiently learning new tasks using previous information is referred to as forward transfer.

Catastrophic forgetting and forward transfer are often thought of as competing desiderata of continual learning where one has to strike a balance between the two depending on the application at hand (Hadsell et al., 2020). Specifically, Wolczyk et al. (2021) recently studied the interplay of forgetting and forward transfer in the robotics context, and found that many continual learning approaches alleviate catastrophic forgetting at the expense of forward transfer. This is indeed unavoidable if the capacity of the model is less than the amount of information we intend to store. However, assuming that the model has sufficient capacity to learn all the tasks simultaneously, as in multitask learning, one might think that a less forgetful model could transfer its retained knowledge to future tasks when they are similar to past ones.

---

*Work done while interning at DeepMind.
†JC and AC contributed equally.

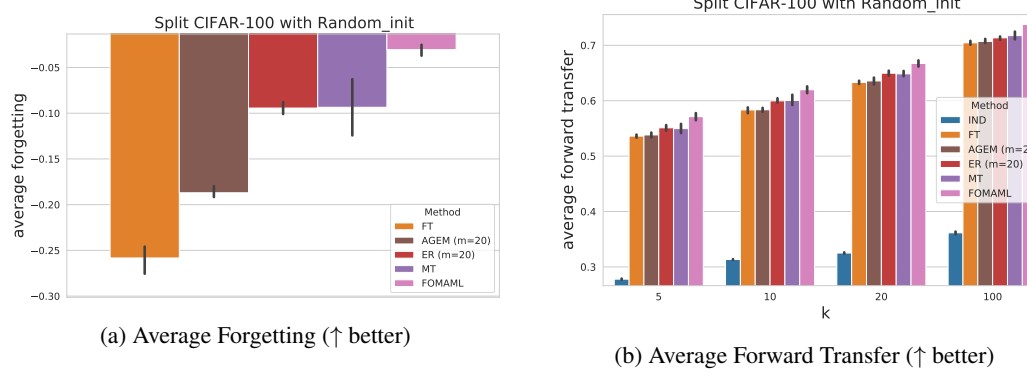

(a) Average Forgetting (↑ better)

(b) Average Forward Transfer (↑ better)

Figure 1: Comparing average forgetting with average forward transfer for different continual learning methods using random initialization on the Split CIFAR-100 benchmark. FOMAML has less forgetting and thus better forward transfer.

In this work, therefore, we argue for looking at the trade-off between forgetting and forward transfer in the right perspective. Typically, forward transfer is measured as the learning accuracy on a task after the continual learner has already made training updates from the task (Wolczyk et al., 2021; Chaudhry et al., 2019a; Lopez-Paz & Ranzato, 2017). However, since such training updates are usually modified to preserve performance on previous tasks (e.g. EWC (Kirkpatrick et al., 2016)), a competition arises between maximizing learning accuracy and mitigating catastrophic forgetting. *Therefore, we argue for a measure of forward transfer that is unconstrained from any training modifications made to preserve previous knowledge.* We propose to use *auxiliary evaluation* of continually trained representations as a measure of forward transfer which is separate from the continual training of the model. Specifically, at the arrival of a new task, we fix the representations learned on the previous task and evaluate them on the new task. This evaluation is done by learning a temporary classifier using a small subset of data from the new task and measuring performance on the test set of the task. The continual training on the new task then proceeds with the updates to the representations (and the classifier) with the full training dataset of the task. We note that this notion of forward transfer removes the tug of war between forgetting the previous tasks and transfer to the next task, and it is with this notion of transfer that we ask the question *are less forgetful representations more transferable?*

We analyze the interplay of catastrophic forgetting and forward transfer on several supervised continual learning benchmarks and algorithms. For this work, we restrict ourselves to the task-based continual learning setting, where task information is assumed at both train and test times as it makes the aforementioned evaluation based on auxiliary classification at fixed points easily interpretable. Our results demonstrate that *a less forgetful model in fact transfers better* (cf. **Figure 1**). We find this observation to be true for both randomly initialized models as well as for models that are initialized from a pre-trained model. We further analyse the reasons of this better transferability and find that less forgetful models result in more diverse and easily separable representations making it easier to learn a classifier head on top. We note that with these results, we want to emphasize that the continual learning community should look at the trade-off between forgetting and forward transfer in the right perspective. The learning accuracy based measure of forward transfer is useful for end-to-end learning on a fixed benchmark and it creates a trade-off between forgetting and forward transfer as rightly demonstrated by Hadsell et al. (2020); Wolczyk et al. (2021). However, in the era of foundation models where pretrain-then-finetune is a dominant paradigm and where one often does not know a priori the tasks where a foundation model will be finetuned, a measure of forward transfer that looks at the capability of a backbone model to be finetuned on several downstream tasks is perhaps a more apt measure.

The rest of the paper is organized as follows. In **Section 2**, we describe the training and evaluation setups considered in this work. In **Section 3**, we provide experimental details followed by the main results of the paper. **Section 4** lists down most relevant works to our study. We conclude with **Section 5** providing some hints to how the findings of this study can be useful for the future research.

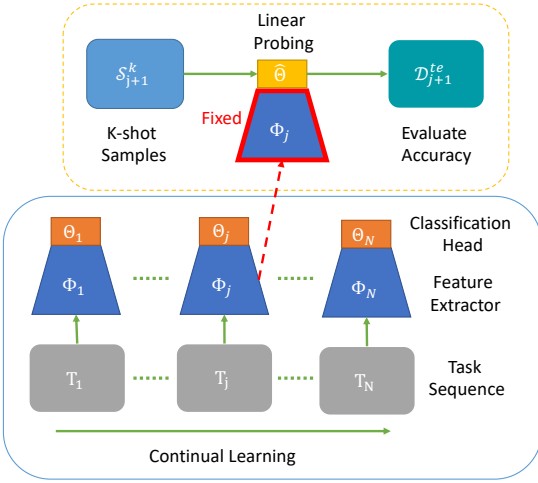

Figure 2: Illustration of continual learning and k-shot evaluation process. We continuously train the feature extractor and the classification head on a task sequence $T_1, \ldots, T_N$. $\Theta_j \circ \Phi_j$ is the model obtained after training on $T_j$. To evaluate the forward transfer of $\Phi_j$, we use linear probing on k-shot samples from the next task $T_{j+1}$ to learn a classifier $\hat{\Theta}$ and then evaluate the accuracy of $\hat{\Theta} \circ \Phi_j$ on the test set $\mathcal{D}_{j+1}^{te}$ from the task $T_{j+1}$.

## 2 PROBLEM SETUP AND METRICS

We consider a supervised continual learning setting consisting of a sequence of tasks $\mathcal{T} = \{T_1, \cdots, T_N\}$. A task $T_j$ is defined by a dataset $\mathcal{D}_j = \{(x_i, y_i, t_i)_{i=1}^{n_j}\}$, consisting of $n_j$ triplets, where $x \in \mathcal{X}$, $y \in \mathcal{Y}$, and $t \in \mathbb{T}$ are input, label and task id, respectively. Each $\mathcal{D}_j = \{\mathcal{D}_j^{tr}, \mathcal{D}_j^{val}, \mathcal{D}_j^{te}\}$ consists of train, validation and test sets. At a given task 'j', the learner may have access to all the previous tasks' datasets $\{\mathcal{D}_i\}_{i<j}$, but it will not have access to the future tasks. We define a feed-forward neural network consisting of a feature extractor $\Phi : \mathcal{X} \mapsto \mathbb{R}^D$ and a task-specific classifier $\Theta_j : \mathbb{R}^D \times \mathbb{T} \mapsto \mathcal{Y}_j$, that implements an input to output mapping $f_j = (\Theta_j \circ \Phi) : \mathcal{X} \times \mathbb{T} \mapsto \mathcal{Y}_j$. The neural network is trained by minimizing a loss $\ell_j : f_j(\mathcal{X}, \mathbb{T}) \times \mathcal{Y}_j \mapsto \mathbb{R}^+$ using stochastic gradient descent (SGD) (Bottou, 2010). While we consider image classification tasks and use cross-entropy loss for each task, the approach would be applicable to other tasks and loss functions as well.

The learner updates a shared feature extractor ($\Phi$) and task-specific heads ($\Theta_j$) throughout the continual learning experience. After training on each task 'i', we measure the performance of the learner on all the tasks observed so far. Let $\text{Acc}(i, j)$ be the accuracy of the model on $\mathcal{D}_j^{te}$ after the feature extractor is updated with $T_i$. We define the average forgetting metric at task 'i' similar to (Lopez-Paz & Ranzato, 2017):

$$\text{Fgt}_i = \frac{1}{i-1} \sum_{j=1}^{i-1} \text{Acc}(i, j) - \text{Acc}(j, j).$$

The average forgetting metric ($\in [-1, 1]$) throughout the continual learning is then defined as,

$$\text{AvgFgt} = \frac{1}{N-1} \sum_{i=2}^{N} \text{Fgt}_i. \tag{1}$$

A negative value of $\text{Fgt}_i$ indicates that the learner has lost performance on the previous tasks, and the more negative $\text{AvgFgt}$ is the more forgetful the representations are of the previous knowledge.

**Forward Transfer through K-Shot Probing** We measure forward transfer in terms of how *easy* it is to learn a new task given continually trained representations. The easiness is measured by learning a linear classifier on top of the *fixed* representations using a small subset of the data of the new task

(refer to **Figure 2** for illustration). Specifically, let $\mathcal{S}_{j+1}^k \overset{k}{\sim} \mathcal{D}_{j+1}^{tr}$ denote a sample consisting of 'k' examples per class from $\mathcal{D}_{j+1}^{tr}$, and let $\Phi_j$ be the representations obtained after training on task 'j' (see the bottom blob of **Figure 2**). Let $\hat{\Theta}$ be the temporary (linear) classifier head learned on top of fixed $\Phi_j$ using $\mathcal{S}_{j+1}^k$. We measure the accuracy of this temporary classifier on the test set of task 'j+1' and denote it as $\texttt{Fwt}_j^k$. This is called the forward transfer of learned representations $\Phi_j$ to the next task 'j+1'. The average forward transfer throughout the continual learning is then defined as,

$$\texttt{AvgFwt}^k = \frac{1}{N-1} \sum_{j=1}^{N-1} \texttt{Fwt}_j^k. \tag{2}$$

We note that linear probing is an auxiliary evaluation process where model updates during evaluation remain distinct from the updates made by the continual learner while observing a task sequence. Contrary to this, in most prior works (Wolczyk et al., 2021; Lopez-Paz & Ranzato, 2017), forward transfer is measured after the continual learner has made updates on the task. Such updates typically restrict the learning on current task to alleviate catastrophic forgetting on the previous tasks. This causes the learner to perform worse on the current task compared to a learner that is not trying to mitigate catastrophic forgetting. We sidestep this dilemma by separating the updates made by the continual learner on a new task from the temporary updates made during auxiliary evaluation on a copy of the model. We also note that similar to linear probing, one could finetune the whole model, including the representations, during the auxiliary evaluation. The main argument is to decouple the notion of forward transfer from modifications made by the continual learning algorithm to preserve knowledge of the previous tasks.

**Feature Diversity**   In addition to $\texttt{AvgFgt}$ (**Equation 1**) and $\texttt{AvgFwt}^k$ (**Equation 2**), we also measure how diverse and easily separable the features of our trained models are for analyzing the transferability of the representations. Specifically, let $\Psi_j \in \mathbb{R}^{m \times D}$ be the feature matrix computed using the feature extractor $\Phi_j$ (obtained after training on task 'j') on the '$m$' test examples of task 'j+1'. Let $\Psi_j^c$ be a sub-matrix constructed by collecting the rows of $\Psi_j$ that belong to class 'c'. Similar to (Wu et al., 2021; Yu et al., 2020), we define the feature diversity score of $\Phi_j$ as

$$\texttt{FDiv}_j = \log |\alpha \Psi_j^\top \Psi_j + \mathbf{I}| - \sum_{c=1}^{C_j} \log |\alpha_j \Psi_j^{c\top} \Psi_j^c + \mathbf{I}|,$$

where $|\cdot|$ is a matrix determinant operator, $\alpha = D/(m\epsilon^2)$, $\alpha_j = D/(m_j\epsilon^2)$, $\epsilon = 0.5$, and $C_j$ denotes the number of classes for task 'j'. The average feature score throughout the continual learning experience is then defined as,

$$\texttt{AvgFDiv} = \frac{1}{N-1} \sum_{j=1}^{N-1} \texttt{FDiv}_j. \tag{3}$$

The intuition behind using this score is that features that enforce high inter-class separation and low intra-class variability should make it easier to learn a classifier head on top leading to a better transfer to next tasks.

## 3 EXPERIMENTS & RESULTS

### 3.1 SETUP

We now briefly describe the experimental setup including the benchmarks, approaches and training details. More details can be found in Appendix A. After the experimental details, we provide the main results of the paper.

**Benchmarks**

- **Split CIFAR-10**: We split CIFAR-10 dataset (Krizhevsky et al., 2009) into 5 disjoint subsets corresponding to 5 tasks. Each task has 2 classes.

- **Split CIFAR-100**: We split CIFAR-100 dataset (Krizhevsky et al., 2009) into 20 disjoint subsets corresponding to 20 tasks. Each task has 5 classes.
- **CIFAR-100 Superclasses**: We split CIFAR-100 dataset into 5 disjoint subsets corresponding to 5 tasks. Each task has 20 classes from 20 superclasses in CIFAR-100 respectively.
- **CLEAR**: This is a continual image classification benchmark by Lin et al. (2021), built from YFCC100M (Thomee et al., 2016) images, containing the evolution of object categories from years 2005-2014. There are 10 tasks each containing images in chronological order from years (2005-2014). We consider both CLEAR10 (consisting of 10 object classes) and CLEAR100 (consisting of 100 object classes) variants of the benchmark.
- **Split ImageNet**: We split ImageNet (Russakovsky et al., 2015) dataset into 100 disjoint subsets corresponding to 100 tasks. Each task has 10 classes.

For all the benchmarks, except split ImageNet, we considered continual learning from a randomly initialized model as well as from a pre-trained ImageNet model. For split ImageNet, we only considered continual learning from a randomly initialized model.

**Approaches**  [1]

Below we describe the approaches considered in this work. Except for the independent baseline, all other baselines reuse the model *i.e.* continue training the same model used for the previous tasks.

- **Independent (IND)**: Trains a model from an initial model (either random initialized or pre-trained) on each task independently.
- **Finetuning (FT)**: Trains a single model on all the tasks in a sequence, one task at a time.
- **Linear-Probing-Finetuning (LP-FT)**: LP-FT (Kumar et al., 2021) is the same as FT except that before each task training, we first learn a task-specific classifier $\Theta$ for the task via linear probing and then train both the feature extractor $\Phi$ and the classifier $\Theta$ on the task to reduce the feature drift.
- **Multitask (MT)**: Trains the model on the data from both the current and previous tasks using the multitask training objective (**equation 6** in Appendix). The data of previous tasks is used as an auxiliary loss while learning on the current task.
- **Experience Replay (ER)**: Uses a replay buffer $\mathcal{M} = \cup_{i=1}^{N-1} \mathcal{M}_i$ when learning on the task sequence, where $\mathcal{M}_i$ stores $m$ examples per class from the task $T_i$. It trains the model on the data from both the current task and the replay buffer when learning on the current task using an ER training objective (**equation 7** in Appendix) (Chaudhry et al., 2019b). There are two main differences between MT and ER: (1) MT uses all the data from the previous tasks while ER only uses limited data from the previous tasks; (2) MT chooses the coefficient for the auxiliary loss via cross-validation while ER always set it to be $1$.
- **AGEM**: Projects the gradient when doing the SGD updates so that the average loss on the data from the episodic memory does not increase (Chaudhry et al., 2019a). The episodic memory stores $m$ examples per class from each task.
- **FOMAML**: First-order MAML (FOMAML) is a meta-learning approach proposed by Finn et al.. We modify FOMAML such that it can be used in the continual learning setting. Similar to MT, FOMAML uses all the data from the previous tasks when learning the current task. The training objective of FOMAML aims to enable knowledge transfer between different batches from the same task. The learning algorithm for FOMAML is provided in Appendix A.2.

**Architecture and Training details.**    We use ResNet50 (He et al., 2016) architecture as the feature extractor $\Phi$ on all benchmarks. On CLEAR10 and CLEAR100, we use a single classification head $\Theta$ that is shared by all the tasks (single-head architecture) while on other benchmarks, we use a separate classification head $\Theta_i$ for each task $T_i$ (multi-head architecture). We use SGD to update the model parameters and use cosine learning rate scheduling (Loshchilov & Hutter, 2016) to adjust the learning rate during training. For LP-FT, we use a base learning rate of $0.001$ while for other

---

[1]The EWC (Kirkpatrick et al., 2016) results are in Appendix **Table 6**.

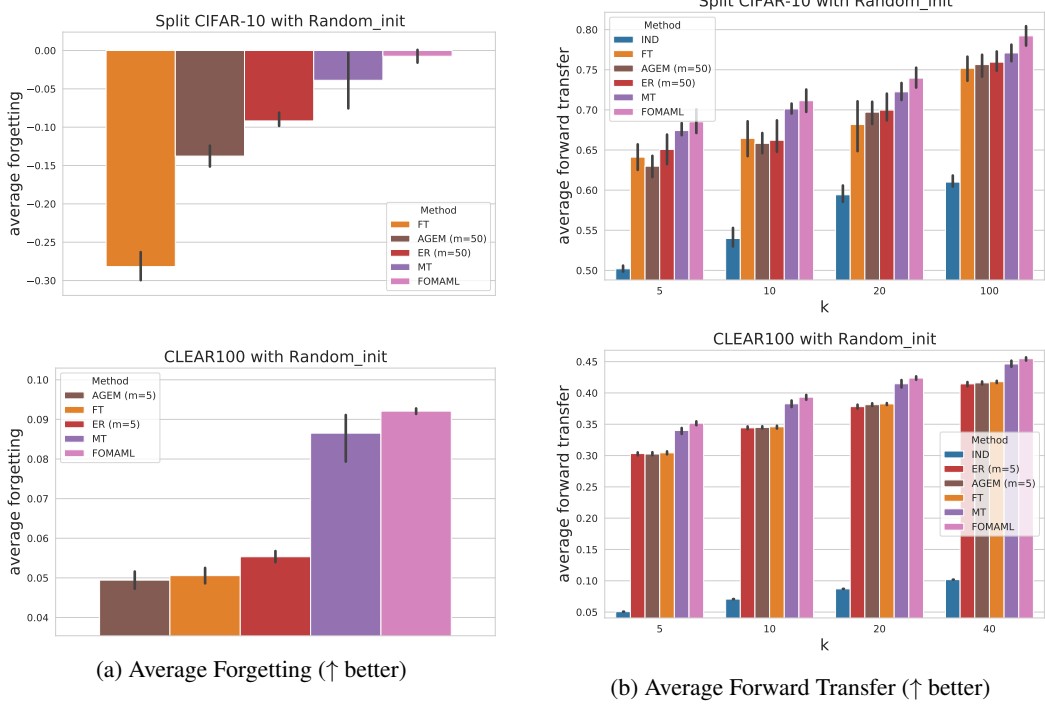

(a) Average Forgetting (↑ better)

(b) Average Forward Transfer (↑ better)

Figure 3: Comparing average forgetting with average forward transfer for different continual learning methods using random initialization on the Split CIFAR-10 and CLEAR100 benchmarks.

baselines, we use a base learning rate of $0.01$. When using a random initialization as the initial model $f_0$, on split CIFAR-10, split CIFAR-100 and CIFAR-100 superclasses, we train the model for 50 epochs per task while on CLEAR10, CLEAR100 and Split ImageNet, we train the model for 100 epochs per task. When using a pre-trained model as the initial model $f_0$, on all benchmarks, we train the model for 20 epochs per task as we found it sufficient for training convergence. For CLEAR10 and CLEAR100, the results are averaged over 5 different runs with different random seeds each corresponding to a different network initialization, where the task order is fixed. For other benchmarks, the results are averaged over 5 different runs, where each run corresponds to a different random ordering of tasks. The results are reported as averages and 95% confidence interval estimates of these 5 runs. For k-shot linear probing, we use SGD with a fixed learning rate of $0.01$. We train the classifier head $\hat{\Theta}$ for 100 epochs on the k-shot dataset $\mathcal{S}_{j+1}^k$ as we found it sufficient for training convergence. On CLEAR100, we consider $k \in \{5, 10, 20, 40\}$ while on other benchmarks, we consider $k \in \{5, 10, 20, 100\}$.

## 3.2 RESULTS

### LESS FORGETFUL REPRESENTATIONS TRANSFER BETTER

We assess the compatibility between forgetting and transferability through `AvgFgt` and `AvgFwt`$^k$ metrics described in **Section 2**. **Figures 1 and 3** show these two metrics for Split CIFAR-100, Split-CIFAR10 and CLEAR100, respectively when the continual learning experience begins from a **randomly initialized model** (the comparison on the other benchmarks is provided in the Appendix B.1). It can be seen from the figures that if a model has less average forgetting, the corresponding model representations have a better K-shot forward transfer. For example, on all the three benchmarks visualized in the figures, FOMAML and MT tend to have the least amount of average forgetting. Consequently, the `AvgFwt`$^k$ of these two baselines is higher compared to all the other baselines, for all the values of $k$ considered in this work. Note that the ranking of other methods in terms of correspondence between forgetting and forward transfer is roughly maintained as well. This shows that *when continual learning experience begins from a randomly initialized model, retaining the knowledge of the past tasks or forgetting less on those tasks is a good inductive bias for forward transfer.*

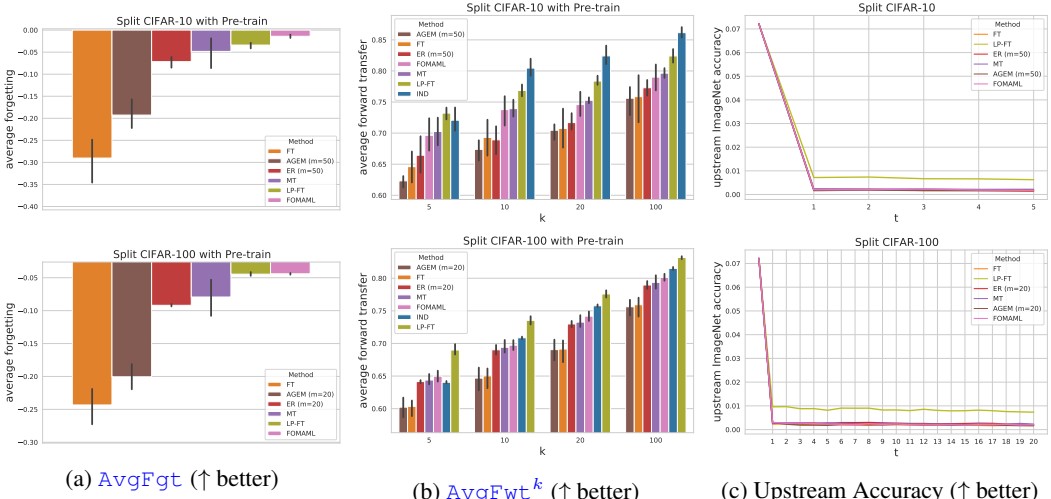

(a) `AvgFgt` (↑ better)  (b) `AvgFwt`$^k$ (↑ better)  (c) Upstream Accuracy (↑ better)

Figure 4: Comparing average forgetting with average forward transfer for different continual learning methods that train the model from a pre-trained ImageNet model on the Split CIFAR-10 and Split CIFAR-100 benchmarks. We also show the accuracy of the models on the upsteam ImageNet data. Since CIFAR-10 and CIFAR-100 images have different image resolution than that of ImageNet images, we need to resize the ImageNet test images from $224 \times 224$ to $32 \times 32$ in order to get meaningful accuracy of the models trained on the Split CIFAR-10 and Split CIFAR-100 benchmarks on the upstream ImageNet data (although the accuracy of the pre-trained model on the resized ImageNet test images is significantly reduced).

Recently, Mehta et al. (2021) showed that pre-trained models tend to forget less, compared to randomly initialized models, when trained on a sequence of tasks. We build upon this observation and ask if forgetting less on both the upstream (pre-trained) task, and downstream tasks improve the transferability of the representations? **Figure 4** shows the comparison between forgetting (left) and forward transfer (middle) on Split CIFAR-10 and Split CIFAR-100 when the continual learning experience begins from a pre-trained model (the comparison on the other benchmarks is provided in the Appendix B.1). It can be seen from the figure that except for LP-FT, less forgetting is a good indicator of a better forward transfer. In order to understand, why LP-FT has a better forward transfer, compared to FOMAML and MT, despite having higher forgetting on the continual learning benchmark at hand, we evaluate the continually updated representations on the upstream data (test set of ImageNet). The evaluation results are given in the right plot of **Figure 4**. From the plot, it can be seen that LP-FT has retained better upstream performance (relatively speaking) compared to the other baselines. This follows our general thesis that *retaining 'previous knowledge', evidenced here by the past performance on both the upstream and downstream tasks, is a good inductive bias for forward transfer.* If instead of freezing the representations and just updating the classifier, we finetune the whole model in the auxiliary evaluation, the less forgetful representations still transfer better (refer to Appendix B.5).

In order to aggregate the metrics across different methods and to see a global trend between forgetting and forward transfer, we compute the Spearman rank correlation between `AvgFgt` and `AvgFwt`$^k$. Table 1 shows the correlation values for different values of 'k' for both randomly initialized and pre-trained models. It can be seen from the table that most of the entries are above 0.5 and statistically significant ($p < 0.01$) showing that reducing forgetting improves the forward transfer across the board.

LESS FORGETFUL REPRESENTATIONS ARE MORE DIVERSE

We now look at what makes the less forgetful representations amenable for better forward transfer. We hypothesize that less forgetful representations maintain more diversity and discrimination in the features making it easy to learn a classifier head on top leading to better forward transfer. To measure this diversity of representations, we look at the feature diversity score `AvgFDiv`, as defined in **Equation 3**, and compare it with the average forgetting score `AvgFgt`. **Table 2** shows the results on different benchmarks and approaches. It can be seen from the table that, for randomly initialized models, less forgetting, evidenced by higher `AvgFgt` score, generally leads to repre-

| Dataset | Random Init | | | Pretrain | | |
|---|---|---|---|---|---|---|
| | $k=5$ | $k=10$ | $k=20$ | $k=5$ | $k=10$ | $k=20$ |
| Split CIFAR-10 | 0.64 | 0.56 | 0.65 | 0.68 | 0.66 | 0.58 |
| Split CIFAR-100 | 0.92 | 0.88 | 0.91 | 0.86 | 0.86 | 0.88 |
| CIFAR100 Superclasses | 0.3 (0.11) | 0.4 (0.03) | 0.4 (0.03) | 0.16 (0.40) | 0.46 (0.01) | 0.62 |
| CLEAR10 | 0.7 | 0.65 | 0.73 | 0.43 (0.02) | 0.37 (0.04) | 0.64 |
| CLEAR100 | 0.58 | 0.59 | 0.53 | 0.83 | 0.8 | 0.81 |
| Split ImageNet | 0.85 | 0.85 | 0.79 | - | - | - |

Table 1: Spearman correlation between `AvgFgt` and `AvgFwt`$^k$ for different $k$, which computes the correlation over different settings (different training methods and random runs). $p$-values are shown in parenthesis if greater than or equal to 0.01.

| Dataset | Method | Random Init | | Pre-trained | |
|---|---|---|---|---|---|
| | | `AvgFgt` ↑ | `AvgFDiv` ↑ | `AvgFgt` ↑ | `AvgFDiv` ↑ |
| Split CIFAR-10 | FT | $-28.18 \pm 2.97$ | $35.59 \pm 10.52$ | $-29.01 \pm 7.97$ | $60.18 \pm 36.35$ |
| | LP-FT | - | - | $-3.39 \pm 1.06$ | $\mathbf{171.41} \pm 13.41$ |
| | ER (m=50) | $-9.18 \pm 1.50$ | $37.33 \pm 14.66$ | $-7.15 \pm 1.97$ | $66.18 \pm 35.74$ |
| | AGEM (m=50) | $-13.77 \pm 2.38$ | $35.79 \pm 16.34$ | $-19.26 \pm 5.01$ | $60.77 \pm 41.80$ |
| | MT | $-3.88 \pm 5.86$ | $36.88 \pm 13.21$ | $-4.83 \pm 5.56$ | $86.88 \pm 21.82$ |
| | FOMAML | $\mathbf{-0.75} \pm 1.39$ | $\mathbf{45.52} \pm 7.82$ | $-1.40 \pm 0.61$ | $65.26 \pm 10.36$ |
| Split CIFAR-100 | FT | $-25.83 \pm 2.43$ | $224.27 \pm 3.63$ | $-24.33 \pm 4.19$ | $263.31 \pm 27.46$ |
| | LP-FT | - | - | $-4.46 \pm 0.46$ | $\mathbf{332.10} \pm 2.97$ |
| | ER (m=20) | $-9.44 \pm 1.11$ | $225.95 \pm 2.38$ | $-9.19 \pm 0.28$ | $281.31 \pm 3.59$ |
| | AGEM (m=20) | $-18.70 \pm 1.00$ | $224.46 \pm 2.93$ | $-20.05 \pm 3.12$ | $260.01 \pm 20.32$ |
| | MT | $-9.35 \pm 4.96$ | $225.33 \pm 4.62$ | $-7.93 \pm 4.04$ | $277.14 \pm 8.31$ |
| | FOMAML | $\mathbf{-3.05} \pm 0.98$ | $225.87 \pm 5.31$ | $-4.40 \pm 0.20$ | $271.56 \pm 7.45$ |
| CIFAR-100 Superclasses | FT | $-14.45 \pm 1.02$ | $458.73 \pm 12.99$ | $-13.51 \pm 0.56$ | $599.29 \pm 13.65$ |
| | LP-FT | - | - | $-2.66 \pm 0.53$ | $\mathbf{702.43} \pm 4.10$ |
| | ER (m=5) | $-11.33 \pm 1.79$ | $463.78 \pm 7.86$ | $-11.36 \pm 1.44$ | $600.23 \pm 23.86$ |
| | AGEM (m=5) | $-12.28 \pm 0.84$ | $459.65 \pm 14.52$ | $-12.11 \pm 0.76$ | $594.70 \pm 27.51$ |
| | MT | $-1.30 \pm 4.02$ | $465.47 \pm 7.84$ | $-5.50 \pm 3.65$ | $601.38 \pm 16.92$ |
| | FOMAML | $\mathbf{1.99} \pm 0.76$ | $\mathbf{470.27} \pm 5.17$ | $-1.24 \pm 0.44$ | $620.66 \pm 10.34$ |
| CLEAR10 | FT | $0.93 \pm 1.01$ | $76.72 \pm 1.70$ | $0.14 \pm 0.42$ | $265.72 \pm 1.08$ |
| | LP-FT | - | - | $0.87 \pm 0.11$ | $\mathbf{281.78} \pm 0.34$ |
| | ER (m=10) | $1.79 \pm 0.24$ | $76.76 \pm 1.62$ | $-0.05 \pm 0.23$ | $263.89 \pm 0.82$ |
| | AGEM (m=10) | $2.03 \pm 0.86$ | $76.00 \pm 1.79$ | $-0.01 \pm 0.19$ | $266.36 \pm 1.02$ |
| | MT | $\mathbf{4.49} \pm 0.99$ | $\mathbf{79.01} \pm 1.41$ | $0.77 \pm 0.51$ | $265.25 \pm 1.49$ |
| | FOMAML | $\mathbf{4.49} \pm 0.56$ | $77.98 \pm 1.27$ | $0.84 \pm 0.34$ | $262.88 \pm 1.28$ |
| CLEAR100 | FT | $5.06 \pm 0.29$ | $179.47 \pm 1.01$ | $-0.03 \pm 0.13$ | $441.22 \pm 0.50$ |
| | LP-FT | - | - | $1.52 \pm 0.07$ | $\mathbf{488.94} \pm 0.71$ |
| | ER (m=5) | $5.53 \pm 0.24$ | $181.39 \pm 1.56$ | $0.34 \pm 0.18$ | $440.48 \pm 0.79$ |
| | AGEM (m=5) | $4.94 \pm 0.36$ | $179.12 \pm 1.03$ | $0.04 \pm 0.14$ | $441.26 \pm 0.27$ |
| | MT | $8.65 \pm 0.96$ | $\mathbf{186.16} \pm 3.31$ | $1.56 \pm 0.15$ | $444.38 \pm 0.41$ |
| | FOMAML | $\mathbf{9.21} \pm 0.12$ | $184.38 \pm 1.58$ | $1.55 \pm 0.16$ | $440.30 \pm 0.71$ |
| Split ImageNet | FT | $-54.62 \pm 2.26$ | $271.98 \pm 0.96$ | - | - |
| | ER (m=10) | $-27.56 \pm 0.63$ | $289.18 \pm 3.99$ | - | - |
| | AGEM (m=10) | $-50.53 \pm 1.58$ | $274.07 \pm 1.20$ | - | - |
| | MT | $-16.79 \pm 0.69$ | $274.88 \pm 2.13$ | - | - |
| | FOMAML | $\mathbf{-10.58} \pm 0.69$ | $\mathbf{305.63} \pm 5.59$ | - | - |

Table 2: Comparing `AvgFgt` with `AvgFDiv`. The numbers for `AvgFgt` are percentages. **Bold** numbers are superior results.

sentations that have higher `AvgFDiv` score. Similarly, on pre-trained models, methods with lower overall forgetting between the upstream and downstream tasks, such as LP-FT, leads to the highest `AvgFDiv` score. These results suggest that less forgetful representations tend to be more diverse and discriminative.

## 4 RELATED WORKS

Continual Learning (also known as Life-long Learning) (Ring, 1995; Thrun, 1995) aims to learn a model on a sequence of tasks that has good performance on all the tasks observed so far. However, SGD training, relying on IID assumption of data, tends to result in a degraded performance on older tasks, when the model is updated on new tasks. This phenomenon is known as catastrophic forget-

ting (McCloskey & Cohen, 1989; Goodfellow et al., 2014) and it has been a main focus of continual learning research. There are several methods that have been proposed to alleviate catastrophic forgetting, ranging from regularization-based approaches (Kirkpatrick et al., 2016; Aljundi et al., 2018; Nguyen et al., 2018; Zenke et al., 2017), to methods based on episodic memory (Lopez-Paz & Ranzato, 2017; Chaudhry et al., 2019a; Aljundi et al., 2019; Hayes et al., 2018; Riemer et al., 2019; Rolnick et al., 2018; Prabhu et al., 2020) to the algorithms based on parameter isolation (Yoon et al., 2018; Mallya & Lazebnik, 2018; Wortsman et al., 2020; Mirzadeh et al., 2021b; Farajtabar et al., 2020). Besides the algorithmic innovations to reduce catastrophic forgetting, recently some works looked at the role of training regimes (Mirzadeh et al., 2020) and network architectures (Mirzadeh et al., 2021a; 2022) for understanding the catastrophic forgetting phenomenon.

While a learner that reduces catastrophic forgetting tries to preserve the knowledge of the past tasks, often what is more important is to utilize the accrued knowledge to learn new tasks more efficiently, a phenomenon known as forward transfer (Lopez-Paz & Ranzato, 2017; Chaudhry et al., 2019a). In most existing works, the forward transfer to a task is measured as the learning accuracy of the task *after training on the task is finished*. Hadsell et al. (2020) and Wolczyk et al. (2021) argued that continual learning methods that avoid catastrophic forgetting do not improve the forward transfer, in fact, sometimes the catastrophic forgetting is reduced at the expense of the forward transfer (as measured by the learning accuracy). This begs the question whether reducing catastrophic forgetting is a good objective for continual learning research or should the community shift focus on the forward transfer as there seems to be a tug of war between the two?

Contrary to previous work, here, we take an auxiliary evaluation perspective to forward transfer where instead of asking whether reducing forgetting on previous tasks, during training on the current task, improves the current task learning, we ask whether a learner that has less forgetting on previous tasks, results in network representations that can quickly be adapted to new tasks? We argue that this mode of measuring forward transfer decouples the notion of transfer from the restricted updates on the current task employed by a continual learner to avoid forgetting on previous tasks. To the best of our knowledge, most similar to our work is Javed & White (2019); Beaulieu et al. (2020) who also looked at the network representations in the context of continual learning. But they took a converse perspective – arguing that learning transferable representations via meta-learning alleviates catastrophic forgetting.

## 5 CONCLUSION

We are interested in understanding how to continuously accrue knowledge for sample efficient learning of downstream tasks. Similar to some previous works, here we question what effect alleviating catastrophic forgetting has on the efficiency of learning new tasks. However, by contrast, we study forward transfer by the auxiliary evaluation of continually trained representations learned through the course of training on a sequence of tasks. To this end, we evaluated several training algorithms on a sequence of tasks and find that our forward transfer metric is highly correlated with the amount of knowledge retention (i.e. less negative forgetting score), indicating that forgetting less may serve as a good inductive bias for forward transfer.

The question of how to accrue knowledge from the past tasks to learn new tasks more efficiently is ever more relevant with the recent advancements in the large scale models trained using internet scale data, aka foundation models, where we would want to avoid initialization from scratch to save computation time. Our suggested measure of forward transfer, that evaluates continually trained representations, also fits nicely in the context of comparing generalization of different large scale models, where a model that can transfer to multiple downstream tasks is preferred. We are in the era of discovering new capabilities of models, as new capabilities emerge with larger scale. The extrapolation of our findings could mean that a less forgetful foundation model – where forgetting is evaluated on the upstream data – should be preferred over a forgetful model, as the former could transfer better to downstream tasks. This serves as a useful model selection mechanism which can be further explored in future research.

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

# Supplementary Material

## Is Forgetting Less a Good Inductive Bias for Forward Transfer?

## A EXPERIMENTAL DETAILS

### A.1 DATASETS

We describe the details of the datasets used in this paper below:

**CIFAR-10.** The CIFAR-10 dataset (Krizhevsky et al., 2009) consists of 60,000 32x32 color images in 10 classes, with 6,000 images per class. There are 50,000 training images and 10,000 test images. We reserve 10,000 training images as the validation data. So the training set used has 40,000 images. We split the CIFAR-10 dataset into 5 disjoint subsets to create the Split CIFAR-10 benchmark. Split CIFAR-10 has 5 tasks corresponding to the 5 disjoint subsets and each task has 2 classes. During training, we apply random cropping and random horizontal flip to the training images.

**CIFAR-100.** The CIFAR-100 dataset (Krizhevsky et al., 2009) is just like the CIFAR-10, except it has 100 classes containing 600 images each. There are 500 training images and 100 testing images per class. The 100 classes in the CIFAR-100 are grouped into 20 superclasses. Each image comes with a "fine" label (the class to which it belongs) and a "coars" label (the superclass to which it belongs). We reserve 10,000 training images as the validation data. So the training set used has 40,000 images. We use the CIFAR-100 dataset to create two benchmarks Split CIFAR-100 and CIFAR-100 Superclasses. The split CIFAR-100 benchmark is created by splitting the CIFAR-100 dataset into 20 disjoint subsets corresponding to 20 tasks. Each task in Split CIFAR-100 has 5 classes. The CIFAR-100 Superclasses benchmark is created by splitting the CIFAR-100 dataset into 5 disjoint subsets corresponding to 5 tasks. Each task in CIFAR-100 Superclasses has 20 classes from 20 superclasses respectively. For both Split CIFAR-100 and CIFAR-100 Superclasses benchmarks, during training, we apply random cropping and random horizontal flipping data augmentations to the training images.

**CLEAR.** CLEAR (Lin et al., 2021) is the first continual image classification benchmark dataset with a natural temporal evolution of visual concepts in the real world that spans a decade (2005-2014). It contains two continual learning benchmark CLEAR10 and CLEAR100. The original CLEAR10 has 33,000 training images and 5,500 test images with 10 tasks and 11 classes (including a BACKGROUND class). We Remove the BACKGROUND class and reserve 5,000 training images as the validation data. So the training set used has 25,000 images and the test set used has 5,000 images. The original CLEAR100 has 99,963 training images and 50,000 test images with 10 tasks and 100 classes. We reserve 19,991 training images as the validation data. So the training set used has 79,972 images. Each task in CLEAR10 (or CLEAR100) contains images from a certain year (2005-2014). For both CLEAR10 and CLEAR100 benchmarks, we resize the images to $224 \times 224$ and use random cropping and random horizontal flipping data augmentations during training.

**ImageNet.** ILSVRC 2012, commonly known as "ImageNet" (Russakovsky et al., 2015) is a large scale image dataset with 1,000 classes organized according to the WordNet hierarchy. It has 1,281,167 training images and 50,000 validation images with labels. It also has 100,000 test images but without labels. We use the validation images as the test set and reserve 300,000 training images (300 images per class) as the validation set. So the training set used contains 981,167 images. We split the ImageNet dataset into 100 disjoint subsets corresponding to 100 tasks. Each task has 10 classes. During training, we use random cropping and random horizontal flipping data augmentations. To reduce computational cost, we resize the images to $64 \times 64$.

### A.2 BASELINES

We consider the following approaches for leaning on a sequence of tasks $T_1, \ldots, T_N$. Except for the independent baseline, all other baselines reuse the model (*i.e.*) continue training the same model

used for the previous tasks. Specifically, when learning on the task $T_i$ ($i > 1$), the continual learning method will use the model $f_{i-1}$ learned on the previous task $T_{i-1}$ as an initialization for the model $f_i$, which we call *model reusing*. For the first task $T_1$ training, the continual learning method will initialize the model by an initial model $f_0$ (either a random intialization or a pre-trained model).

**Independent (IND).** The IND baseline learns on each task independently. That is when learning on the task $T_i$, it will train the model from an initial model (either a random initialization or a pre-trained model) using the Empirical Risk Minimization (ERM) objective:

$$\min_{f_i} \mathbb{E}_{(x,y,t)\sim\mathcal{D}_i} \ell_i(f_i(x,t), y) \tag{4}$$

We use the IND baseline as a reference to see how well we can learn on a task without learning on other tasks in the task sequence.

**Finetuning (FT).** Finetuning is a simple baseline for continual learning. It trains a single model on all the tasks in a sequence. When training the model $f_i$ on the current task $T_i$, it doesn't use the data from the previous tasks. It uses the ERM objective (**equation 4**) to train the model $f_i$ on the current task $T_i$ only.

**Linear-Probing-Finetuning (LP-FT).** Linear-Probing-Finetuning (Kumar et al., 2021) is the same as the Finetuning baseline except that before each task training, we first learn a task-specific classifier $\Theta$ for the task via linear probing and then train both the feature extractor $\Phi$ and the classifier $\Theta$ on the task. That is when learning on the current task $T_i$, we have a two-stage training process. In the first stage, we train the classifier $\Theta_i$ while fixing the feature extractor $\Phi_i$ via the ERM training objective:

$$\min_{\Theta_i} \mathbb{E}_{(x,y,t)\sim\mathcal{D}_i} \ell_i(f_i(x,t), y; \Theta_i, \Phi_i) \tag{5}$$

In the second stage, we train both the classifier $\Theta_i$ and the feature extractor $\Phi_i$ via the ERM training objective (**equation 4**). We only use LP-FT in the setting where the initial model $f_0$ is a pre-trained model.

**Multitask (MT).** The Multitask baseline trains the model on the data from both the current task and the previous tasks when learning on the current task $T_i$. It uses the following multitask training objective:

$$\min_{\Phi_i, \{\Theta_j\}_{j=1}^{i}} \mathbb{E}_{(x,y,t)\sim\mathcal{D}_i} \ell_i(f_i(x,t), y; \Theta_i, \Phi_i) + \lambda \cdot \mathbb{E}_{1 \le j < i} \mathbb{E}_{(x,y,t)\sim\mathcal{D}_j} \ell_j(f_j(x,t), y; \Theta_j, \Phi_i) \tag{6}$$

It trains a shared feature extractor $\Phi_i$ and task-specific classifiers $\{\Theta_j\}_{j=1}^{i}$ for the tasks $T_1, \ldots, T_i$. The hyperparameter $\lambda$ is chosen from the set $\{1.0, 0.1, 0.01\}$ based on the average learning accuracy across tasks on the validation set.

**Experience Replay (ER).** The ER baseline uses a replay buffer $\mathcal{M} = \cup_{i=1}^{N-1} \mathcal{M}_i$ when learning on the sequence of tasks, where $\mathcal{M}_i$ stores examples from the task $T_i$. In our work, we restrict the replay buffer $\mathcal{M}$ to store only $m$ examples per class from each task. The training objective used by ER is:

$$\min_{\Phi_i, \{\Theta_j\}_{j=1}^{i}} \mathbb{E}_{(x,y,t)\sim\mathcal{D}_i} \ell_i(f_i(x,t), y; \Theta_i, \Phi_i) + \mathbb{E}_{1 \le j < i} \mathbb{E}_{(x,y,t)\sim\mathcal{M}_j} \ell_j(f_j(x,t), y; \Theta_j, \Phi_i) \tag{7}$$

**AGEM.** AGEM is a continual learning method proposed by Chaudhry et al.. Similar to ER, AGEM also uses a replay buffer (or an episodic memory) $\mathcal{M} = \cup_{i=1}^{N-1} \mathcal{M}_i$, where $\mathcal{M}_i$ stores only $m$ examples per class from the task $T_i$. While learning the task $T_i$, the training objective of AGEM is:

$$\min_{f_i} \quad \mathbb{E}_{(x,y,t)\sim\mathcal{D}_i}\ell_i(f_i(x,t),y) \tag{8}$$

$$\text{s.t.} \quad \mathbb{E}_{(x,y,t)\sim\mathcal{M}_1^{i-1}}\ell_t(f_i(x,t),y) \leq \mathbb{E}_{(x,y,t)\sim\mathcal{M}_1^{i-1}}\ell_t(f_{i-1}(x,t),y) \tag{9}$$

where $\mathcal{M}_1^{i-1} = \cup_{j=1}^{i-1}\mathcal{M}_j$. The corresponding optimization problem is:

$$\min_{\tilde{g}} \frac{1}{2}\|g - \tilde{g}\|_2^2 \quad \text{s.t.} \quad \tilde{g}^T g_{ref} \geq 0 \tag{10}$$

where $g$ is a gradient computed using a batch randomly sampled from the current task to solve the objective (**equation 8**), $g_{ref}$ is a gradient computed using a batch randomly sampled from the episodic memory $\mathcal{M}_1^{i-1}$, and $\tilde{g}$ is a projected gradient that we will use to update the model. When the gradient $g$ violates the constraint (**equation 9**), it is projected via:

$$\tilde{g} = g - \frac{g^T g_{ref}}{g_{ref}^T g_{ref}} g_{ref} \tag{11}$$

**FOMAML.** MAML is a meta-learning approach proposed by Finn et al.. Since there are some differences in the meta-learning setting and the continual learning setting (e.g. meta-learning algorithms assume that there is a task distribution where we can sample tasks from it while in the continual learning setting, we don't have such a task distribution), we cannot directly use the MAML algorithm proposed in Finn et al. (2017). We then modify MAML such that it can be used in the continual learning setting. While learning on the task $T_i$, the training objective we want to solve is:

$$\min_{f_i} \mathbb{E}_{B\sim\mathcal{D}_i}\ell_i(B;f_i) + \lambda \cdot \mathbb{E}_{j\in[i]}\mathbb{E}_{B_{j,1}^{\text{in}},\ldots,B_{j,b}^{\text{in}},B_j^{\text{out}}\sim\mathcal{D}_j}\ell_j(B_j^{\text{out}};f_{i,j}^{(b)}) \tag{12}$$

where $[i] = \{1, 2, \ldots, i\}$, $B \sim \mathcal{D}_i$ means sampling a batch $B$ from $\mathcal{D}_i$ and

$$f_{i,j}^{(b)} = U_b(B_{j,1}^{\text{in}}, \ldots, B_{j,b}^{\text{in}}; f_i) \tag{13}$$

Here, $U_b(B_1, \ldots, B_b; f)$ is a model obtained by applying $b$ gradient update steps on the model $f$ using $b$ batches $B_1, \ldots, B_b$. If we use standard SGD for $U_b$ and the learning rate is $\alpha$, then we have

$$f_{i,j}^{(0)} = f_i, \quad f_{i,j}^{(q)} = f_{i,j}^{(q-1)} - \alpha \cdot \nabla_{f_{i,j}^{(q-1)}}\ell_j(B_{j,q}^{\text{in}}; f_{i,j}^{(q-1)}), \quad q = 1, \ldots, b \tag{14}$$

The training objective aims to find a model such that it achieves small error on the current task $T_i$ and after several gradient update steps on some batches from a seen task $T_j$ ($j \in [i]$), the updated model can achieve small error on other batches from the task $T_j$. So we want to find a model that can enable knowledge transfer between different batches from the same task.

Solving the objective (**equation 12**) requires computing the second-order gradients, which might be expensive. However, we can use the idea of first-order MAML (FOMAML) proposed in Finn et al. (2017), which ignores the second derivative terms, to solve the objective. The algorithm of FOMAML is presented in Algorithm 1. In our experiments, we simply set $\alpha = \beta$ and $c = 1$. On Split ImageNet, we set $b = 1$ while on other benchmarks, we set $b = 2$.

### A.3 ARCHITECTURE AND TRAINING DETAILS

**Architecture.** We use ResNet50 (He et al., 2016) architecture as the feature extractor $\Phi$ on all benchmarks by default. On CLEAR10 and CLEAR100, we use a single classification head $\Theta$ that is shared by all the tasks (single-head architecture) while on other benchmarks, we use a separate classification head $\Theta_i$ for each task $T_i$ (multi-head architecture).

---

**Algorithm 1** FIRST-ORDER MAML (FOMAML)

---

**Require:** A model $f_{i-1}$ after training on the previous task $T_{i-1}$, a learning rate $\alpha$ for inner-update, a learning rate $\beta$ for outer-update, the number of previous tasks $c$ used for each training step, the number of gradient update steps $b$ for the inner-update and the number of training steps $n$ for the outer-update.

1: $f_i \leftarrow f_{i-1}$
2: Randomly sample a batch $B$ from the current task $T_i$, i.e., $B \sim \mathcal{D}_i$
3: $G \leftarrow \nabla_{f_i} \ell_i(B; f_i)$
4: **for** $p = 1, 2, \ldots, n$ **do**
5:     Randomly select $c$ indices from the set $\{1, 2, \ldots, i-1\}$ without replacement as a set $I_p$.
6:     $I \leftarrow I_p \cup \{i\}$
7:     **for** $j \in I$ **do**
8:        $f_{i,j} \leftarrow f_i$
9:        **for** $q = 1, 2, \ldots, b$ **do**
10:          Randomly sample a batch $B_{j,q}^{\text{in}}$ from $\mathcal{D}_j$.
11:          Apply an inner-update step: $f_{i,j}^{(q)} \leftarrow f_{i,j}^{(q-1)} - \alpha \cdot \nabla_{f_{i,j}^{(q-1)}} \ell_j(B_{j,q}^{\text{in}}; f_{i,j}^{(q-1)})$.
12:        **end for**
13:        Randomly sample a batch $B_j^{\text{out}}$ from $D_j$.
14:        $G \leftarrow G + \nabla_{f_{i,j}^{(q)}} \ell_j(B_j^{\text{out}}; f_{i,j}^{(q)})$
15:     **end for**
16:     Apply an outer-update step: $f_i \leftarrow f_i - \beta \cdot G$
17: **end for**
18: **return** $f_i$.

---

**Continual Learning Training Details.** We use Stochastic Gradient Decent (SGD) for training models. We use cosine learning rate scheduling (Loshchilov & Hutter, 2016) to adjust the learning rate during training. Suppose the base learning rate is $r$ and the number of training steps for each task is $n$. Then for each task training, at training step $t$, the learning rate for the SGD update is $r \cdot \cos \frac{t\pi}{2n}$. For LP-FT, we use a base learning rate of $0.001$ while for other baselines, we use a base learning rate of $0.01$. on split CIFAR-10, split CIFAR-100 and CIFAR-100 superclasses, we use a batch size of $64$. On CLEAR10 and CLEAR100, we use a batch size of $128$. On Split ImageNet, we use a batch size of $256$. When using a random initialization as the initial model $f_0$, on split CIFAR-10, split CIFAR-100 and CIFAR-100 superclasses, we train the model for $50$ epochs per task while on CLEAR10, CLEAR100 and Split ImageNet, we train the model for $100$ epochs per task. When using a pre-trained model as the initial model $f_0$, on all benchmarks, we train the model for $20$ epochs per task as we found it sufficient for training convergence. For LP-FT, we perform the linear probing for $10$ epochs per task. These training hyper-parameters are chosen based on the average learning accuracy across tasks on the validation set.

**K-Shot Linear Probing Training Details.** We use Stochastic Gradient Decent (SGD) with a fixed learning rate of $0.01$ for linear probing. We train the classifier head $\hat{\Theta}$ for $100$ epochs on the k-shot dataset $\mathcal{S}_{j+1}^k$ as we found it sufficient for training convergence. On CLEAR100, the set of values we consider for $k$ is $\{5, 10, 20, 40\}$ while on other benchmarks, the set of values we consider for $k$ is $\{5, 10, 20, 100\}$. For each $k$, we use a batch size of $\min(k \cdot c, 50)$, where $c$ is the number of classes in the task. We don't apply any data augmentations to the training images during the k-shot linear probing.

### A.4 HYPER-PARAMETERS SELECTION

In this section, we discuss how we select the hyper-parameters.

**Continual Learning Training.** For different baselines, the shared hyper-parameters are the batch size, the learning rate and the number of training epochs. We do not tune the batch size, but set it to be a fixed number for each benchmark. For the learning rate and the number of training epochs, we choose them based on the average learning accuracy across tasks on the validation data. The range of the learning rate that we consider is $\{0.1, 0.01, 0.001, 0.0001\}$. We found that setting the learning rate to be $0.01$ leads to the best average learning accuracy for all the baselines except LP-FT. For LP-FT, we found that setting the learning rate to be $0.001$ leads to better average learning accuracy.

We set the number of training epochs to be a sufficiently large number such that the average learning accuracy doesn't improve as we increase the number of epochs further. For all the baselines, we pick a fixed number of training epochs such that all methods converge for each benchmark setting.

**K-shot Linear Probing Training.** The hyper-parameters are the batch size, the learning rate and the number of training epochs. For each $k$, we just set the batch size to be $\min(k \cdot c, 50)$ and do not tune it. Note that K-shot linear probing is a convex optimization problem. Thus, the number of training epochs will not affect the results across baselines as long as we train for a sufficient number of epochs. We found that 100 epochs were more than sufficient for all the baselines to converge for K-shot linear probing. Also, the learning rate will not affect the results much as long as we pick a reasonable one. Therefore, we simply fix the learning rate to be 0.01.

# B ADDITIONAL EXPERIMENTAL RESULTS

## B.1 COMPARING FORGETTING AND FORWARD TRANSFER

In the main paper, we give results for comparing forgetting and forward transfer on some benchmarks. In this section, we provide additional results for comparing forgetting and forward transfer on other benchmarks. **Figure 5** shows the results where we use a random initialization and **Figure 6** shows the results where we use a pre-trained model. We can see that the claims made in **Section 3.2** still hold here.

## B.2 EVALUATING AVERAGE ACCURACY AND AVERAGE LEARNING ACCURACY

In this section, we report results for traditional continual learning metrics Average Accuracy and Average Learning Accuracy. The Average Accuracy is defined as:

$$\texttt{AvgAcc} = \frac{1}{N} \sum_{j=1}^{N} \texttt{Acc}(N, j).$$

While the Average Learning Accuracy is defined as:

$$\texttt{AvgLAcc} = \frac{1}{N} \sum_{j=1}^{N} \texttt{Acc}(j, j).$$

The results are reported in **Table 3**.

## B.3 ABLATION STUDY ON THE MODEL ARCHITECTURE

We want to see whether our claims about forgetting and forward transfer also hold when we use a different architecture. Thus, on split CIFAR-10, split CIFAR-100, and CIFAR-100 Superclasses benchmarks, we also report results in **Figure 7** for using ResNet18 as the model architecture. We only show results for using random initialization since we don't have pre-trained ResNet18 model on ImageNet. From the results, we can see that our claim that less forgetting is a good inductive bias for forward transfer still holds.

## B.4 CORRELATION BETWEEN AVERAGE FORGETTING AND AVERAGE FEATURE DIVERSITY SCORE

In order to aggregate the metrics across different approaches and to see a global trend between forgetting and feature diversity, we compute the Spearman rank correlation between the `AvgFgt` and `AvgFDiv`. Table 4 shows the correlation values for randomly initialized models. From the results, we can see that for randomly initialized models, `AvgFgt` and `AvgFDiv` generally have positive correlations.

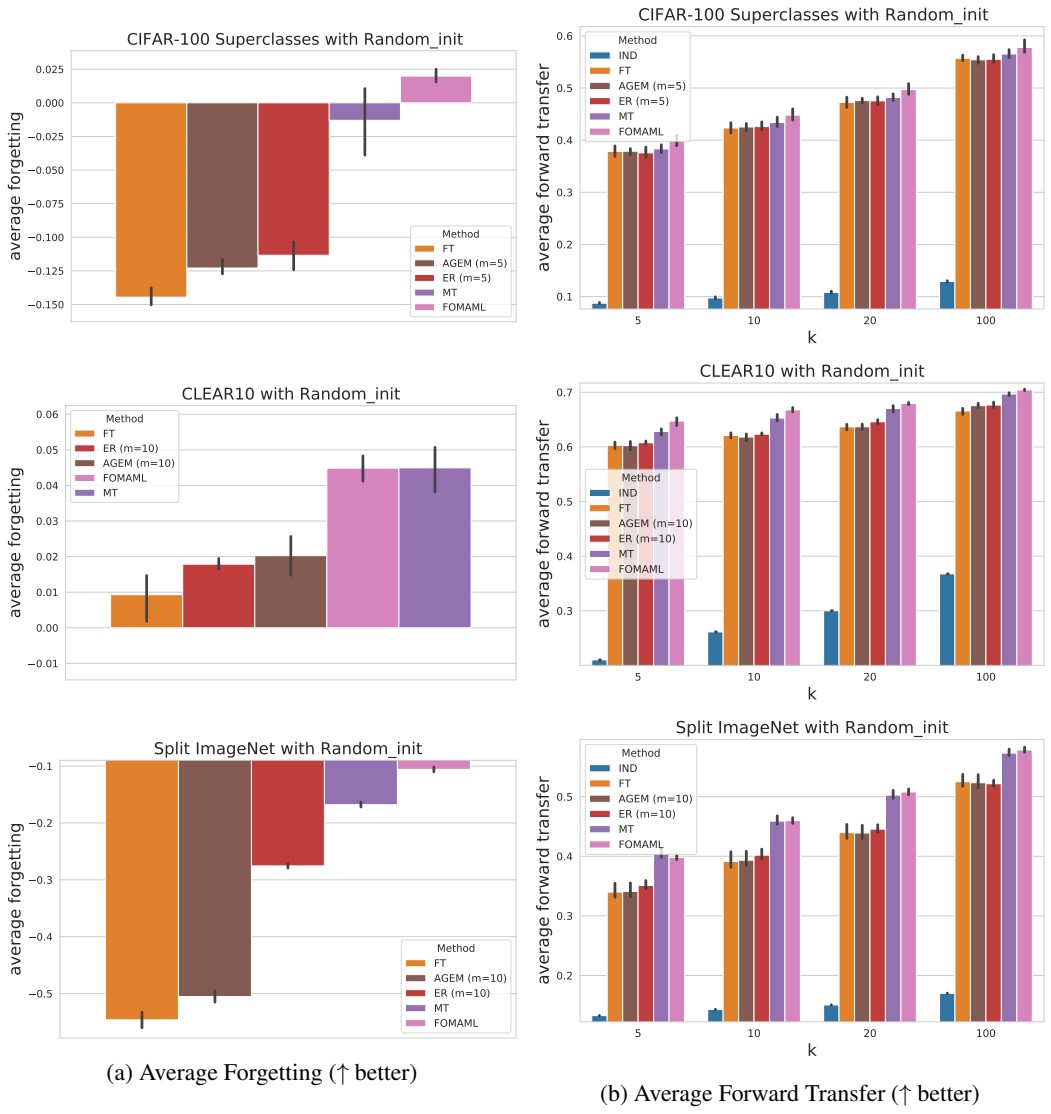

(a) Average Forgetting (↑ better)

(b) Average Forward Transfer (↑ better)

Figure 5: Comparing average forgetting with average forward transfer for different continual learning methods using random initialization on the CIFAR-100 Superclasses, CLEAR10 and Split ImageNet benchmarks.

### B.5 FORWARD TRANSFER THROUGH K-SHOT FINE-TUNING

We also evaluate forward transfer through k-shot fine-tuning (i.e., we fine-tune the entire model including the feature extractor $\Phi_j$ and the classifier $\hat{\Theta}$ on the k-shot samples $\mathcal{S}_{j+1}^k$). The training hyper-parameters are the same as those of k-shot linear probing, except that to avoid overfitting while fine-tuning the whole network, we perform cross-validation for the learning rate and the number of training epochs using the validation set. The learning rate is chosen from the set $\{0.01, 0.001\}$ while the number of training epochs is chosen from the set $\{10, 50, 100\}$. When using random initialization, the results on the Split CIFAR-10, Split CIFAR-100, and CIFAR-100 Superclasses benchmarks are shown in **Figure 8** while the results on the CLEAR10, CLEAR100 and Split ImageNet benchmarks are shown in **Figure 9**. When using a pre-trained model as initialization, the results on the Split CIFAR-10, Split CIFAR-100, CIFAR-100 Superclasses, CLEAR10 and CLEAR100 benchmarks are shown in **Figure 10**.

In order to aggregate the metrics across different approaches and to see a global trend between forgetting and forward transfer, we compute the Spearman rank correlation between the `AvgFgt` and `AvgFwt`$^k$ for the k-shot fine-tuning evaluation. Table 5 shows the correlation values for different

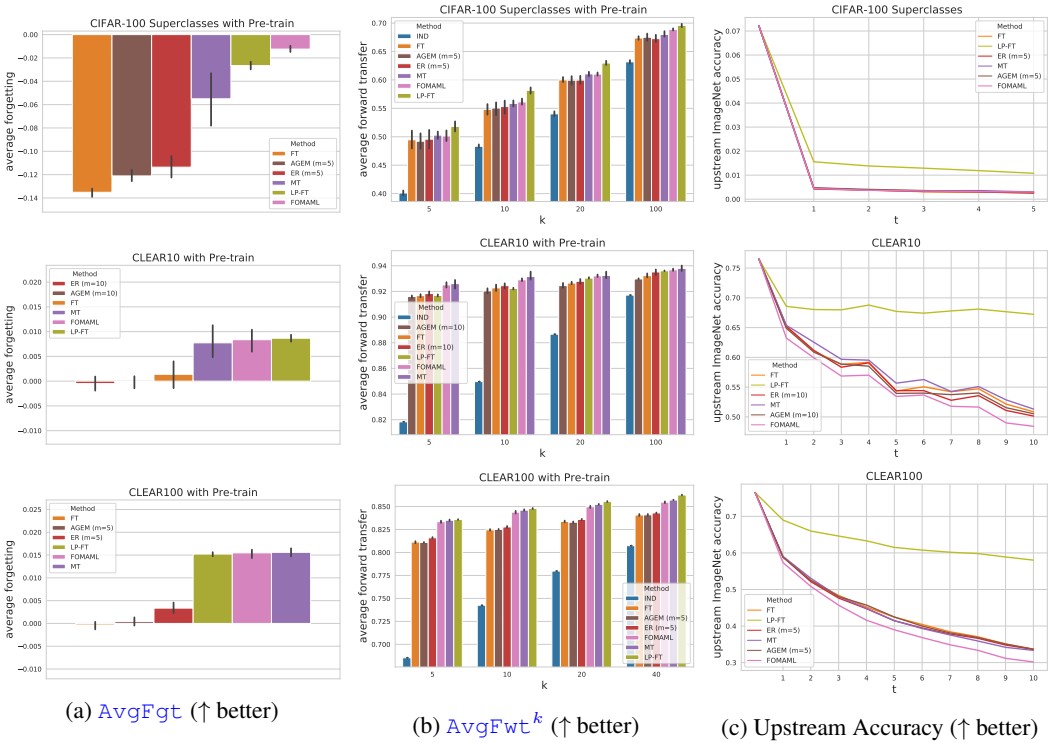

(a) `AvgFgt` (↑ better)  (b) `AvgFwt`$^k$ (↑ better)  (c) Upstream Accuracy (↑ better)

Figure 6: Comparing average forgetting with average forward transfer for different continual learning methods that train the model from a pre-trained ImageNet model on the CIFAR-100 Superclasses, CLEAR10 and CLEAR100 benchmarks. We also show the accuracy of the models on the upsteam ImageNet data. Since CIFAR-100 images have different image resolution than that of ImageNet images, we need to resize the ImageNet test images from $224 \times 224$ to $32 \times 32$ in order to get meaningful accuracy of the models trained on the CIFAR-100 Superclasses benchmark on the upstream ImageNet data. On CLEAR10 and CLEAR100 benchmarks, since their images have the same image resolution as the ImageNet images, we don't need to resize the ImageNet test images when evaluating the upstream accuracy.

values of 'k' for both randomly initialized and pre-trained models. It can be seen from the table that most of the entries are above $0.5$ and statistically significant ($p < 0.01$) showing that reducing forgetting improves the forward transfer across the board.

From the results, we can see that less forgetting generally leads to better forward transfer. Thus, our claim that less forgetting is a good inductive bias for forward transfer still holds.

## B.6    Ablation Study on the replay buffer size

For the Experience Replay (ER) baseline, we perform experiments on the Split CIFAR-100 and CIFAR-100 Superclasses benchmarks to study the effect of the replay buffer size on the `AvgFgt` and `AvgFwt`$^k$ metrics. The results when using random initialization are shown in **Figure 11** while the results when using a pre-trained model as initialization are shown in **Figure 12**. From the results, we can see that increasing $m$ usually leads to less forgetting and thus more forward transfer. Therefore, our claim that less forgetting is a good inductive bias for forward transfer still holds.

## B.7    Results for EWC and Vanilla L2 Regularization

In this section, we provide some results for the EWC method (Kirkpatrick et al., 2016) and the vanilla L2 regularization (a variant of EWC where the fisher information matrix is replaced with an identity matrix) using ResNet18 as the model architecture with random initialization on the Split CIFAR-10 benchmark. For $\lambda$ in EWC, we consider the range $\{10, 50, 100, 200\}$ and select the best one based on the performance on the validation data. For $\lambda$ in vanilla L2 regularization, we consider the range $\{10, 1, 0.1, 0.01\}$ and select the best one based on the performance on the validation

| Dataset | Method | Random Init | | Pretrain | |
|---|---|---|---|---|---|
| | | `AvgAcc` | `AvgLAcc` | `AvgAcc` | `AvgLAcc` |
| Split CIFAR-10 | FT | $62.69 \pm 5.24$ | $91.38 \pm 1.44$ | $67.56 \pm 14.62$ | $95.75 \pm 1.02$ |
| | LP-FT | - | - | $92.64 \pm 1.15$ | $95.46 \pm 0.13$ |
| | ER (m=50) | $85.56 \pm 2.46$ | $91.70 \pm 1.53$ | $90.41 \pm 1.26$ | $95.90 \pm 0.35$ |
| | AGEM (m=50) | $81.74 \pm 3.14$ | $91.59 \pm 1.14$ | $80.22 \pm 5.51$ | $95.88 \pm 1.09$ |
| | MT | $89.70 \pm 5.37$ | $92.68 \pm 0.80$ | $91.48 \pm 4.32$ | $96.14 \pm 0.08$ |
| | FOMAML | $92.77 \pm 0.40$ | $93.45 \pm 0.71$ | $94.61 \pm 0.50$ | $96.06 \pm 0.45$ |
| Split CIFAR-100 | FT | $57.47 \pm 3.11$ | $81.31 \pm 0.93$ | $67.96 \pm 1.90$ | $89.64 \pm 0.79$ |
| | LP-FT | - | - | $85.33 \pm 0.59$ | $90.27 \pm 0.32$ |
| | ER (m=20) | $72.91 \pm 1.19$ | $81.00 \pm 0.42$ | $81.51 \pm 1.00$ | $89.82 \pm 0.20$ |
| | AGEM (m=20) | $64.02 \pm 0.84$ | $81.35 \pm 0.75$ | $71.77 \pm 1.59$ | $89.24 \pm 1.34$ |
| | MT | $73.90 \pm 5.08$ | $82.15 \pm 0.36$ | $82.74 \pm 5.11$ | $90.33 \pm 0.38$ |
| | FOMAML | $79.79 \pm 0.57$ | $82.36 \pm 0.57$ | $85.36 \pm 1.13$ | $89.63 \pm 0.64$ |
| CIFAR100 Superclasses | FT | $56.76 \pm 0.94$ | $68.92 \pm 0.78$ | $70.25 \pm 0.78$ | $81.55 \pm 0.38$ |
| | LP-FT | - | - | $78.88 \pm 0.47$ | $81.22 \pm 0.53$ |
| | ER (m=5) | $60.43 \pm 2.37$ | $68.63 \pm 1.60$ | $72.43 \pm 0.99$ | $81.36 \pm 0.25$ |
| | AGEM (m=5) | $59.28 \pm 0.52$ | $69.42 \pm 0.67$ | $71.79 \pm 0.79$ | $81.60 \pm 0.25$ |
| | MT | $68.43 \pm 3.50$ | $69.54 \pm 1.37$ | $77.28 \pm 3.19$ | $81.77 \pm 0.18$ |
| | FOMAML | $71.48 \pm 1.82$ | $69.86 \pm 2.00$ | $80.11 \pm 0.45$ | $81.03 \pm 0.26$ |
| CLEAR10 | FT | $71.18 \pm 0.42$ | $70.19 \pm 0.19$ | $93.64 \pm 0.48$ | $93.87 \pm 0.21$ |
| | LP-FT | - | - | $95.27 \pm 0.05$ | $94.70 \pm 0.11$ |
| | ER (m=10) | $72.92 \pm 0.63$ | $71.16 \pm 0.24$ | $94.12 \pm 0.22$ | $94.15 \pm 0.27$ |
| | AGEM (m=10) | $71.76 \pm 0.67$ | $70.22 \pm 0.75$ | $93.81 \pm 0.27$ | $94.00 \pm 0.13$ |
| | MT | $77.68 \pm 0.68$ | $73.69 \pm 0.54$ | $95.04 \pm 0.64$ | $94.53 \pm 0.26$ |
| | FOMAML | $78.51 \pm 0.47$ | $74.41 \pm 0.47$ | $95.16 \pm 0.22$ | $94.63 \pm 0.24$ |
| CLEAR100 | FT | $52.38 \pm 0.21$ | $47.27 \pm 0.25$ | $86.47 \pm 0.10$ | $86.29 \pm 0.12$ |
| | LP-FT | - | - | $89.56 \pm 0.04$ | $88.22 \pm 0.06$ |
| | ER (m=5) | $53.16 \pm 0.47$ | $46.99 \pm 0.40$ | $87.25 \pm 0.18$ | $86.49 \pm 0.07$ |
| | AGEM (m=5) | $52.35 \pm 0.25$ | $47.25 \pm 0.41$ | $86.56 \pm 0.16$ | $86.26 \pm 0.18$ |
| | MT | $59.58 \pm 1.56$ | $50.94 \pm 0.55$ | $89.56 \pm 0.12$ | $87.96 \pm 0.13$ |
| | FOMAML | $60.99 \pm 0.54$ | $52.09 \pm 0.67$ | $89.42 \pm 0.12$ | $87.78 \pm 0.06$ |
| Split ImageNet | FT | $13.42 \pm 0.54$ | $72.19 \pm 0.41$ | - | - |
| | ER (m=10) | $43.42 \pm 3.50$ | $69.71 \pm 0.40$ | - | - |
| | AGEM (m=10) | $16.82 \pm 1.30$ | $71.71 \pm 0.47$ | - | - |
| | MT | $54.68 \pm 4.13$ | $73.58 \pm 0.50$ | - | - |
| | FOMAML | $59.73 \pm 4.58$ | $71.85 \pm 1.28$ | - | - |

Table 3: Results for average accuracy `AvgAcc` and average learning accuracy `AvgLAcc`. All numbers are percentages.

| Dataset | Spearman Correlation |
|---|---|
| Split CIFAR-10 | 0.33 (0.10) |
| Split CIFAR-100 | 0.31 (0.13) |
| CIFAR100 Superclasses | 0.38 (0.06) |
| CLEAR10 | 0.46 (0.02) |
| CLEAR100 | 0.80 |
| Split ImageNet | 0.81 |

Table 4: Spearman correlation between `AvgFgt` and `AvgFDiv`, which computes the correlation over different settings (different training methods and random runs). Here, we use random initialization as the initial model. $p$-values are shown in parenthesis if greater than or equal to $0.01$.

data. Experimenting with EWC on larger models and longer benchmarks is computationally very expensive. The comparison of EWC with FT and vanilla L2 regularization is given in **Table 6**. It can be seen from the table that less forgetting leads to better forward transfer. Thus, our claim that less forgetting is a good inductive bias for forward transfer still holds.

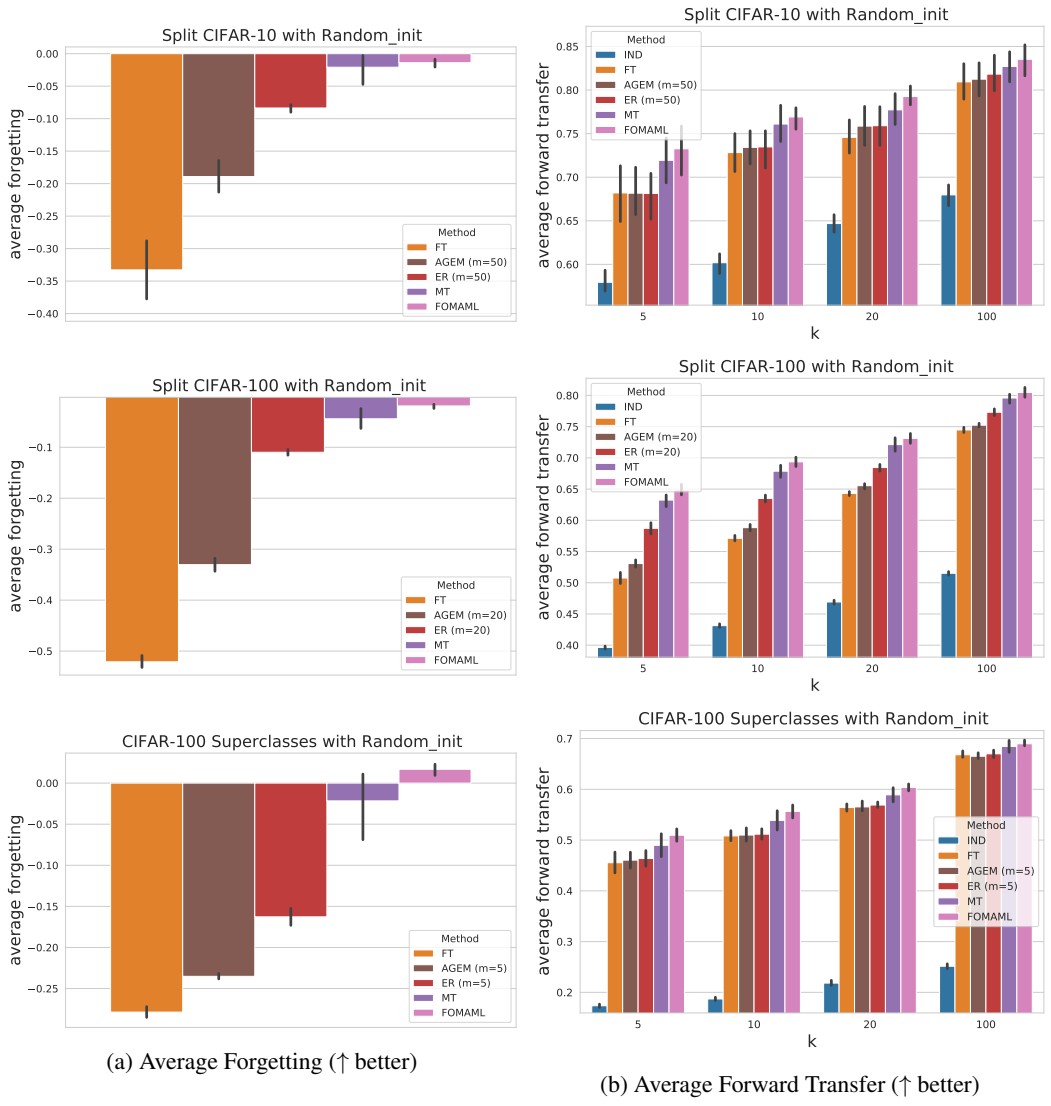

(a) Average Forgetting (↑ better)

(b) Average Forward Transfer (↑ better)

Figure 7: Comparing average forgetting with average forward transfer for different continual learning methods using random initialization on the Split CIFAR-10, Split CIFAR-100 and CIFAR-100 Superclasses benchmarks. Here, we use ResNet18 as the model architecture.

## C  DISCUSSION

### C.1  RELATEDNESS OF TASKS AND OUR CONCLUSIONS

We note here that task relatedness bears significant effect on the relationship between forgetting and forward transfer. The benchmarks that we considered in this work either have very similar tasks (CLEAR10/100), where the same classes are observed over a 10 years period, or unrelated tasks (Split CIFAR10/100, ImageNet), where disjoint classes are observed in each task. In both cases, less forgetting improved the forward transfer, although for more similar tasks the improvement is more significant as intuitively expected. We did not observe that "unrelatedness" of tasks leads to negative transfer. However, if tasks were negatively related to begin with then less forgetting would intuitively lead to negative transfer. But in our experience negatively related tasks are very rare and in practical machine learning systems many tasks can learn from each other (which is the basis of transfer learning, multitask learning, etc.). We would like to emphasize that the point of the paper is precisely to show that when tasks are somewhat related, and observed in a continual setting, less forgetting improves forward transfer. It is in this setting that some of the previous

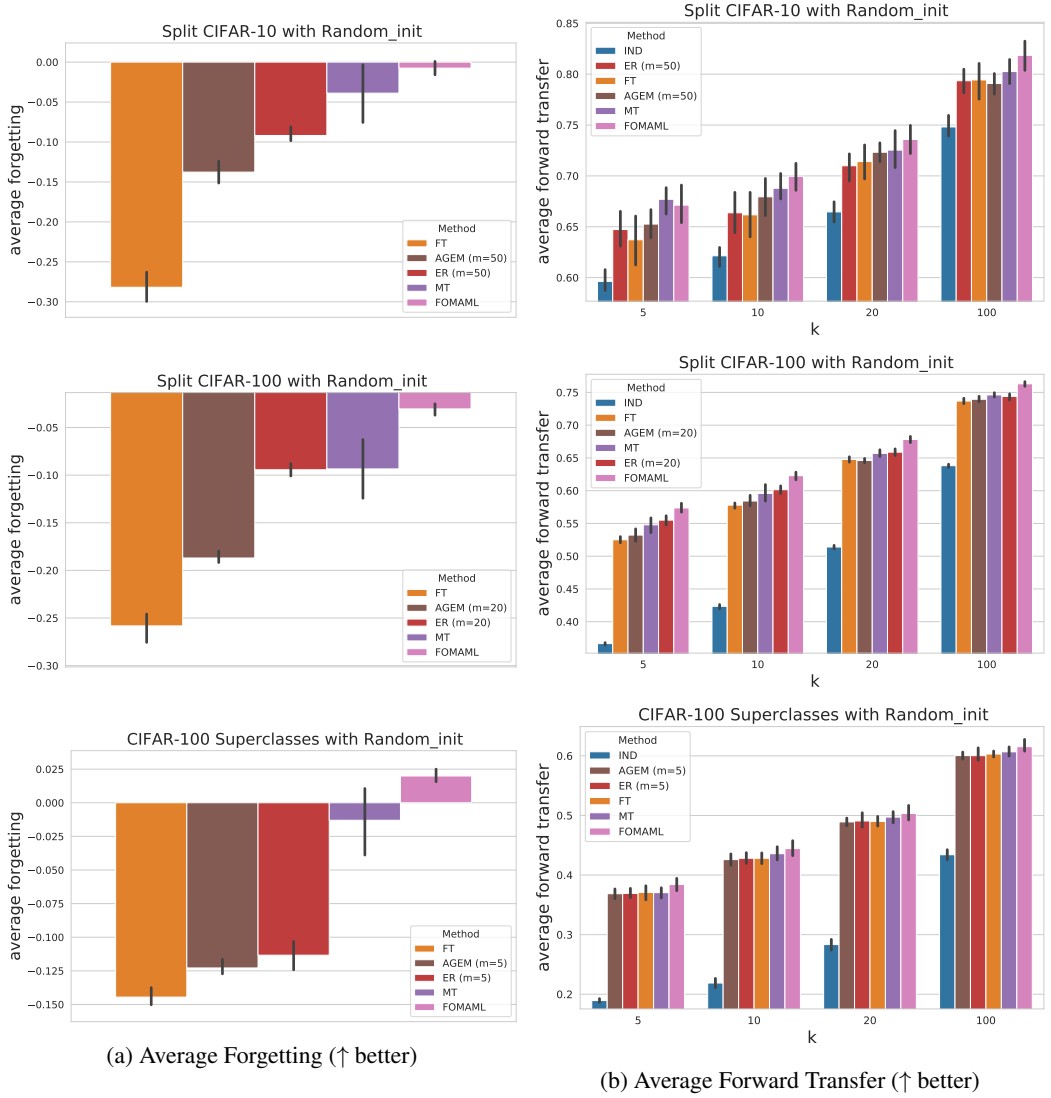

(a) Average Forgetting (↑ better)

(b) Average Forward Transfer (↑ better)

Figure 8: Comparing average forgetting with average forward transfer for different continual learning methods using random initialization on the Split CIFAR-10, Split CIFAR-100 and CIFAR-100 Superclasses benchmarks. Here, we evaluate the forward transfer through k-shot fine-tuning.

works (Hadsell et al., 2020; Wolczyk et al., 2021) concluded that less forgetting does not improve end-to-end forward transfer. We show here that even on such tasks less forgetting improves the representational measure of forward transfer.

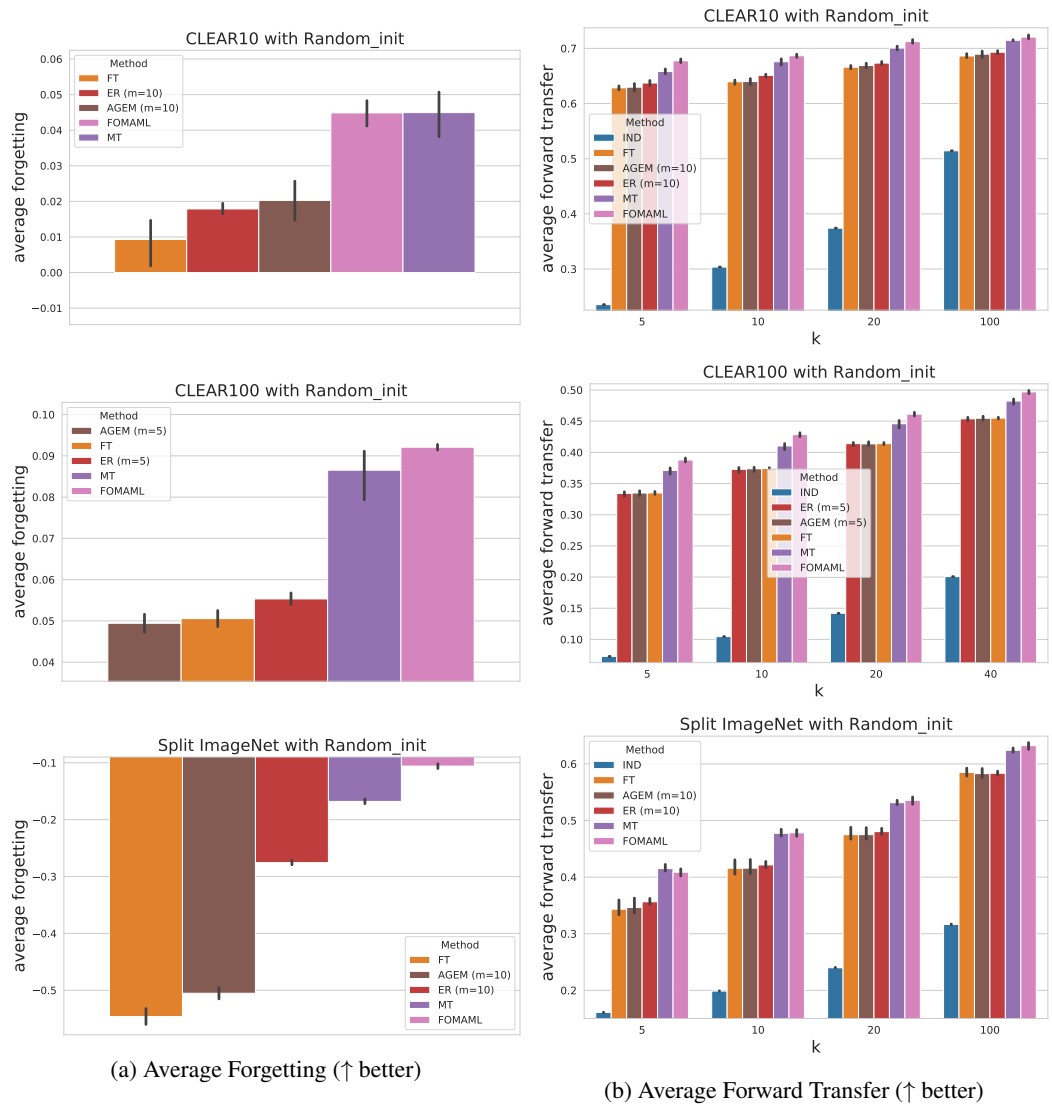

(a) Average Forgetting (↑ better)

(b) Average Forward Transfer (↑ better)

Figure 9: Comparing average forgetting with average forward transfer for different continual learning methods using random initialization on the CLEAR10, CLEAR100 and Split ImageNet benchmarks. Here, we evaluate the forward transfer through k-shot fine-tuning.

| Dataset | Random Init | | | Pretrain | | |
|---|---|---|---|---|---|---|
| | $k = 5$ | $k = 10$ | $k = 20$ | $k = 5$ | $k = 10$ | $k = 20$ |
| Split CIFAR-10 | 0.49 | 0.5 | 0.49 | 0.64 | 0.57 | 0.51 |
| Split CIFAR-100 | 0.92 | 0.93 | 0.83 | 0.87 | 0.88 | 0.86 |
| CIFAR100 Superclasses | 0.16 (0.40) | 0.24 (0.20) | 0.28 (0.14) | 0.28 (0.13) | 0.33 (0.07) | 0.38 (0.04) |
| CLEAR10 | 0.68 | 0.68 | 0.68 | 0.18 (0.34) | 0.25 (0.18) | 0.53 |
| CLEAR100 | 0.6 | 0.59 | 0.61 | 0.87 | 0.87 | 0.83 |
| Split ImageNet | 0.86 | 0.81 | 0.83 | - | - | - |

Table 5: Spearman correlation between `AvgFgt` and `AvgFwt`$_*^k$ for different $k$, which computes the correlation over different settings (different training methods and random runs). Here, `AvgFwt`$_*^k$ is defined like `AvgFwt`$^k$, but instead of using k-shot linear probing for evaluation, we use k-shot fine-tuning evaluation (i.e., fine-tuning the entire model). $p$-values are shown in parenthesis if greater than or equal to $0.01$.

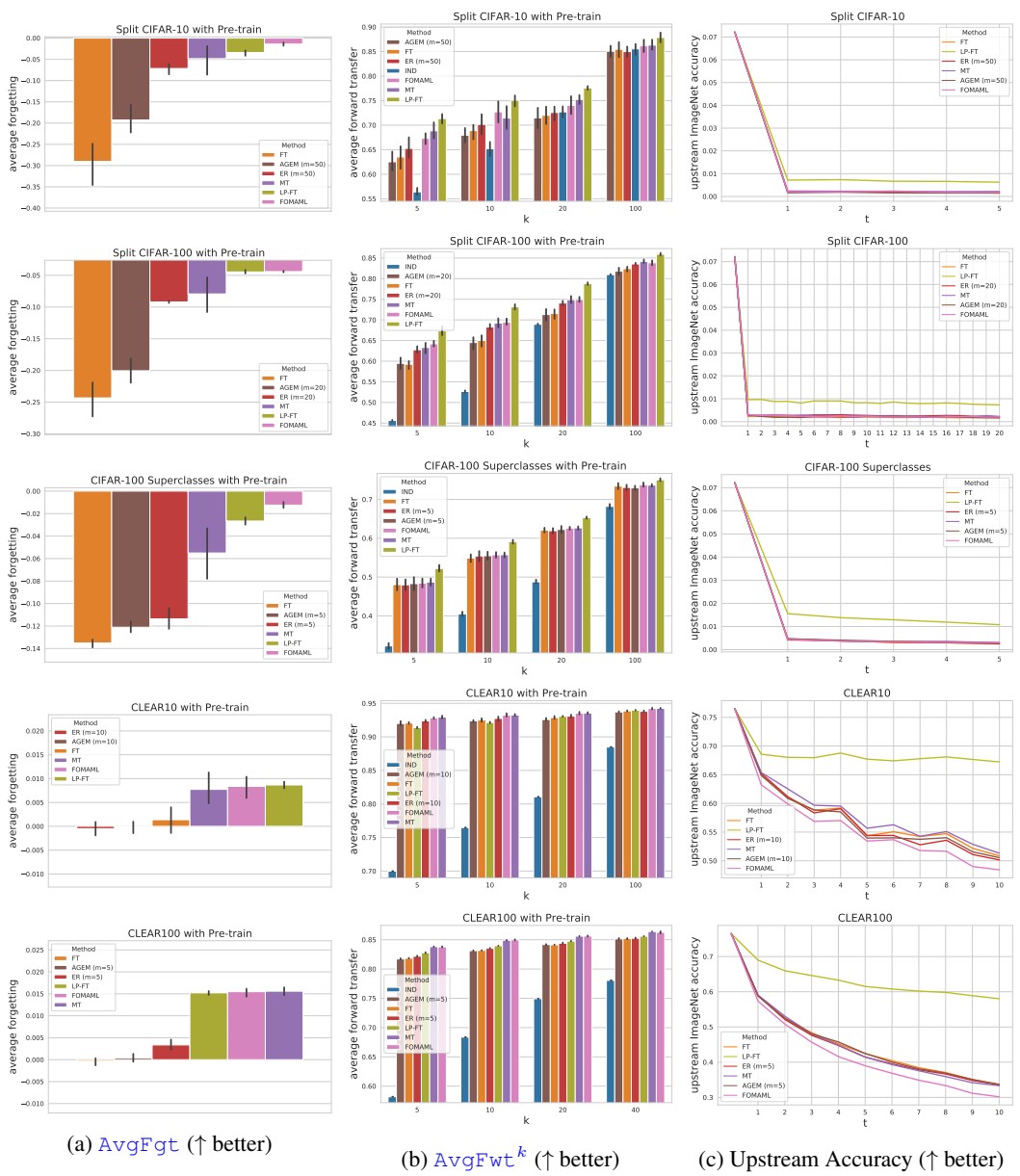

(a) AvgFgt (↑ better)  (b) AvgFwt$^k$ (↑ better)  (c) Upstream Accuracy (↑ better)

Figure 10: Comparing average forgetting with average forward transfer for different continual learning methods that train the model from a pre-trained ImageNet model on the Split CIFAR-10, Split CIFAR-100, CIFAR-100 Superclasses, CLEAR10 and CLEAR100 benchmarks. We also show the accuracy of the models on the upsteam ImageNet data. Here, we evaluate the forward transfer through k-shot fine-tuning.

| Method | AvgFgt ↑ | AvgFwt$^k$ (k=10) | AvgFwt$^k$ (k=20) | AvgFwt$^k$ (k=100) |
|---|---|---|---|---|
| FT | $-32.06 \pm 4.27$ | $71.57 \pm 1.87$ | $74.89 \pm 2.72$ | $79.14 \pm 2.06$ |
| Vanilla L2 Reg. ($\lambda = 0.01$) | $-26.41 \pm 3.02$ | $72.68 \pm 2.28$ | $76.58 \pm 2.73$ | $80.88 \pm 1.92$ |
| EWC ($\lambda = 100$) | $-21.99 \pm 3.51$ | $74.27 \pm 2.09$ | $77.00 \pm 2.03$ | $81.98 \pm 0.93$ |

Table 6: Results for EWC and Vanilla L2 Regularization using ResNet18 as the model architecture with random initialization on the Split CIFAR-10 benchmark. $\lambda$ is a hyper-parameter that controls the regularization strength of EWC and Vanilla L2 Regularization. The numbers are percentages.

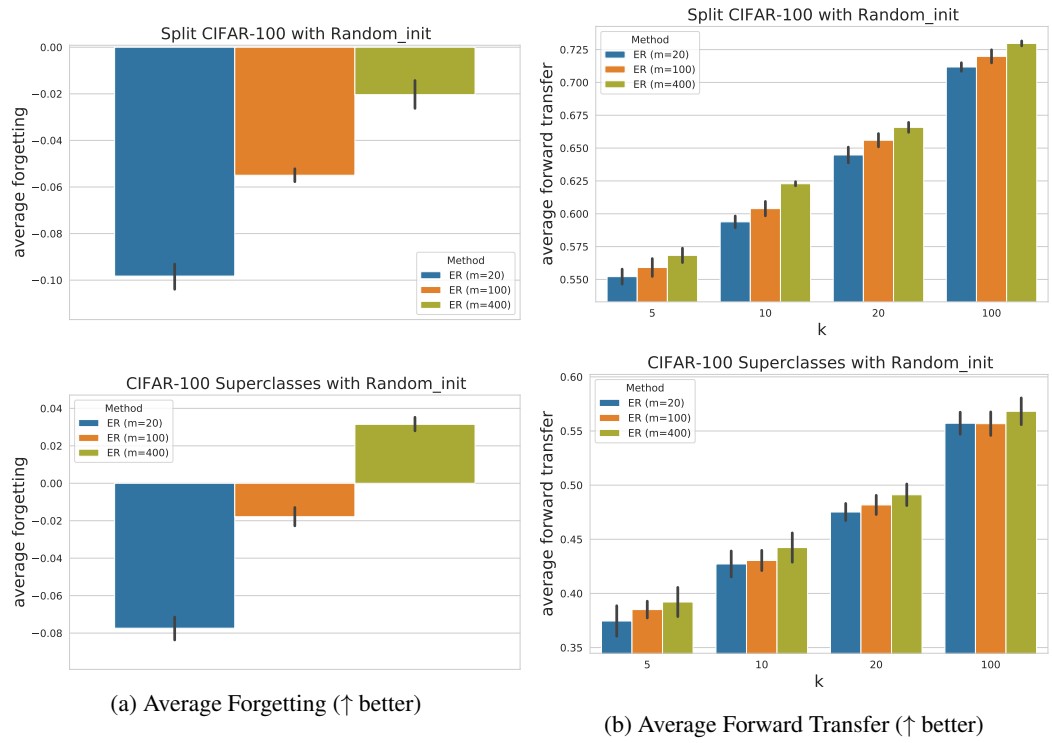

(a) Average Forgetting (↑ better)

(b) Average Forward Transfer (↑ better)

Figure 11: Comparing average forgetting with average forward transfer for the ER method with different replay buffer sizes using random initialization on the Split CIFAR-100 and CIFAR-100 Superclasses benchmarks.

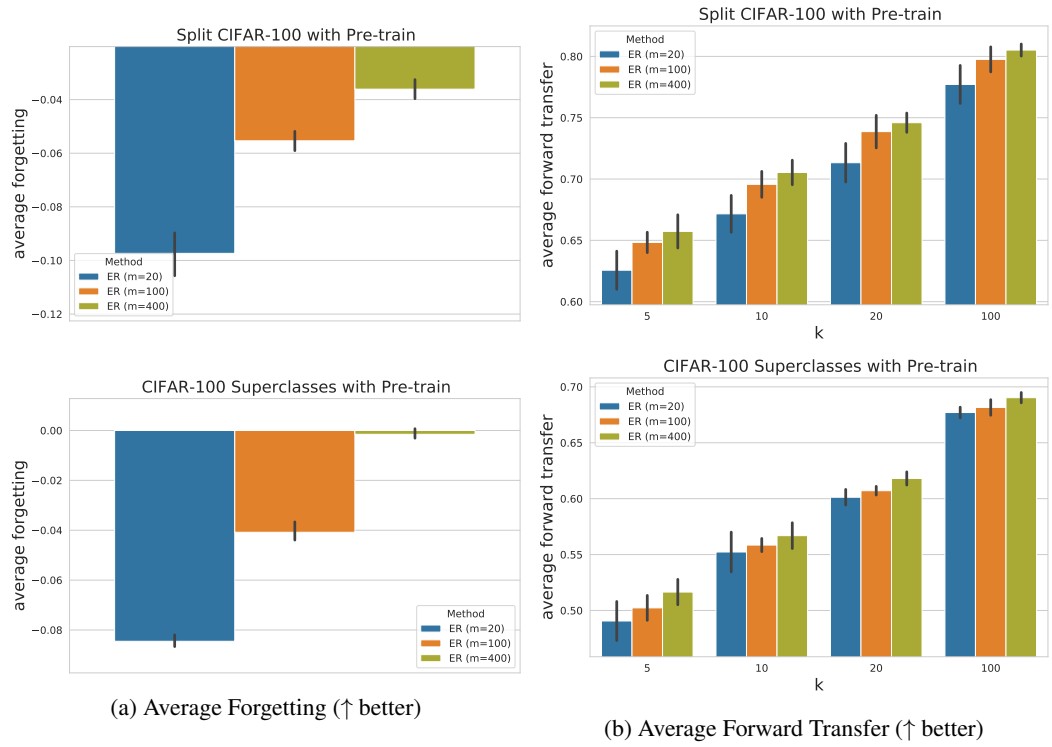

(a) Average Forgetting (↑ better)

(b) Average Forward Transfer (↑ better)

Figure 12: Comparing average forgetting with average forward transfer for the ER method with different replay buffer sizes that trains the model from a pre-trained ImageNet model on the Split CIFAR-100 and CIFAR-100 Superclasses benchmarks.

