# OpenReview forum: "Is Forgetting Less a Good Inductive Bias for Forward Transfer?"
_ICLR.cc/2023/Conference — ICLR 2023 poster_

### Official Review · Reviewer_gCd6 · 2022-10-23

**Confidence:** 4
**Clarity, Quality, Novelty And Reproducibility:** 1. **Clarity.** The paper is clear an…
**Correctness:** 2
**Technical Novelty And Significance:** 2
**Empirical Novelty And Significance:** 3
**Recommendation:** 8

**Details Of Ethics Concerns:**

None.

**Strength And Weaknesses:**

# Strengths
1. **Importance of the research question.** The area of continual learning (CL) is rapidly growing and many *heuristic* methods are proposed to mitigate forgetting and solve continual settings. Thus, I find it important to understand the interplay between different quantities of interest, and specifically, between the forgetting and the forward transfer.
The paper challenges common definitions in CL and proposes a new perspective on transferability, yielding interesting results.

---

# Weaknesses

1. **Proposed transferability notion - Motivation, comparison to existing literature and discussion.**
     While I agree that the proposed evaluation method (using linear probing while fixing the learned representations) makes sense and is *indeed* interesting, I would expect a clearer motivation to be given for it than the one in page 4. There is also place for more discussion.
     Why should one prefer this notion of transferability over the one proposed in [Wolczyk et al. 2021]? Why is an auxiliary-linear-probing even interesting when continual learning algorithms are expected to yield a *single* model?
     Moreover, [Wolczyk et al. 2021] showed that common CL algorithms do not help to (their definition of) forward transfer, and the "preserved" knowledge does not help learning future tasks, either faster (their Figure 3) or better (their Figure 4). In that case, how come these algorithms help linear probing (as seen in the new paper's experiments)? This is a little surprising for me.

1. **Experimental methodology.**

     1. **Some important CL approaches were not tested.** Specifically, no regularization method (e.g., EWC) or parameter-isolation method (e.g., PackNet) were used. I believe these are more important in the paper's context than meta-learning or multitask approaches (FOMAML and MT).

     1. **It is important to report the average accuracy as well.** The average forgetting and forward transfer cannot tell the whole story.

     1. **There are more immediate/natural experiments to conduct.**
     The main statement (Section 3.2) is more comparative in nature (i.e., "less forgetful representations transfer better") than the one in Wolczyk et al., 2021 (i.e., popular CL methods mitigate forgetting but do not contribute to transferability).
     That being said, I feel like comparing *different* CL methods, like done here, can be methodologically problematic (e.g., perhaps LP-FT simply yields better representations that are better for *any* cause, forgetting and transferability included).
     The existing experiments *are* interesting, but I believe simpler experiments could be more convincing: Fix the CL method (e.g., rehearsal/regularization), but test the influence of its main hyperparameter (e.g., buffer-size/regularization-strength, respectively) on forgetting and forward transfer.

     1. **Statistical significance of results.** Even though the authors plot the error-bars according to *one* standard deviation (if I am not mistaken), the results for CIFAR-10 are often not statistically significant (see figures 1 and 3). In contrast, for CIFAR-100 the reported results are more siginificant. Consider performing additional repetitions.

**Summary Of The Paper:**

The paper empirically revisits a previous hypothesis/belief from the continual learning (CL) literature, that catastrophic forgetting is not the only important quantity in CL, and that often CL methods that mitigate forgetting can harm the "forward transfer" between tasks.

The paper offers a new notion of transferability, by proposing an alternative evaluation method: after training a task, one *freezes* the representation learned so far, and uses it for training a $k$-shot linear probing model on the *next* consecutive task.

Using this alternative evaluation method, the authors experimentally show a positive correlation between low-forgetting and high-transferability, when comparing several different continual/multitask-learning algorithms. Finally, lower forgetting is shown to be somewhat correlated to more diverse representations.

**Summary Of The Review:**

The research question is important and interesting. However, it calls for a better motivation and a better comparison to existing work.
The experiments *do* show interesting phenomena, but the methodology lacks some important aspects like I wrote above.

---

> ### Author Response · Authors · 2022-11-11
> **Response to Reviewer gCd6 - Part 1**
>
> We thank the reviewer for their feedback. While we address the main concerns / questions of the reviewer below, we would first like to add a meta-comment clarifying the intent of the paper.
>
> One of the main reasons catastrophic forgetting has received so much attention in the continual learning community is the intuition that a model that forgets less on the past tasks accrues more knowledge and should transfer better to new tasks. This intuition is challenged by some recent works. The purpose of our work is to make sure that the continual learning community looks at the tradeoff between forgetting and forward transfer in the right perspective. In this work, we showed that if looked in the right perspective, forgetting less indeed is a good indicator of the forward transfer. We do not claim that previous studies were wrong. We merely say that they looked at the tradeoff in a different perspective which might be useful for end-to-end learning on a fixed benchmark. But in the era of foundation models where models will follow the pretrain-then-finetune paradigm, looking at the forward transfer from continual representation learning perspective, that we are arguing for, is perhaps a better lens. We have made that point more explicit in the updated draft (all the changes are colored in blue). We hope this sets the context for the reviewer to read our rebuttal.
>
> We now address the individual comments:
>
> 1. **Proposed transferability notion - Motivation, comparison to existing literature and discussion**: We have updated the text to, hopefully, provide a more clear motivation of our proposed notion of forward transfer. As highlighted in the text, the classical notion of forward transfer based on learning accuracy by [Wolczyk et al. 2021 and others] is more suitable when continual learning is done end-to-end on a fixed benchmark and one knows what future tasks to train on. In the era of foundation models, where pretrain-then-finetune is a more dominant paradigm and one may not know in advance where a foundation model will be finetuned, our measure that looks at the transferability of the representations provides a better notion of forward transfer.
>
> &nbsp;&nbsp;&nbsp;&nbsp;&nbsp;&nbsp; The reason [Wolczyk et al. 2021] observed "preserved" knowledge did not help in learning future tasks was how they measured forward transfer. As argued in our paper, they measure forward transfer in terms of learning accuracy which creates a tug of war between how the forgetting is mitigated and how the new task is learned. For example, EWC would regularize the learning of new tasks in order to preserve the knowledge of previous tasks. If the forward transfer were to be measured in terms of the learning accuracy of a new task, then it is not surprising that reducing forgetting by restricting learning on a new task may not improve the forward transfer.
>
> &nbsp;&nbsp;&nbsp;&nbsp;&nbsp;&nbsp; On the other hand, our auxiliary evaluation decouples the notion of forward transfer from modifications made by the continual learning algorithm to preserve knowledge of the previous tasks. In doing so, it gives a better notion of forward transfer of "learned representations".

---

> ### Author Response · Authors · 2022-11-11
> **Response to Reviewer gCd6 - Part 2**
>
>
> 2. **Experimental methodology**:
>
> &nbsp;&nbsp;&nbsp;&nbsp;&nbsp;&nbsp; - **EWC**: We have added the EWC baseline to our experiments. We perform experiments on Split CIFAR-10 using ResNet18 as the model architecture with random initialization. Experimenting with EWC on larger models and longer benchmarks is computationally very expensive. The comparison of EWC with FT is given in the table below. It can be seen from the table that less forgetting leads to better forward transfer. Thus, our main conclusions hold on the regularization-based continual learning approaches as well. We have added these results in the paper (Appendix B.7).
>
>
> | Method           | average forgetting | average forward transfer (k=10) | average forward transfer (k=20) | average forward transfer (k=100) |
> |------------------------|--------------------|---------------------------------|---------------------------------|----------------------------------|
> | FT               |  -32.06 $\pm$ 4.27  | 71.57 $\pm$ 1.87                | 74.89 $\pm$ 2.72                | 79.14 $\pm$ 2.06                 |
> | EWC ($\lambda=100$)   |  -21.99 $\pm$ 3.51  | 74.27 $\pm$ 2.09                | 77.00 $\pm$ 2.03                | 81.98 $\pm$ 0.93                 |
>
>
> &nbsp;&nbsp;&nbsp;&nbsp;&nbsp;&nbsp; - **Average accuracy**: Average accuracy is already reported in the Appendix Table 3. Since our focus was primarily on the forgetting and forward transfer we kept those numbers from the main paper. They don’t change the conclusions of our paper.
>
> &nbsp;&nbsp;&nbsp;&nbsp;&nbsp;&nbsp; - **There are more immediate/natural experiments to conduct**: Upon reviewer’s suggestion, we conducted experiments where we fixed the continual learning algorithm (Experience Replay in this case) and increased the replay buffer size. We conducted these experiments on Split CIFAR-100 and CIFAR-100 superclasses both with random initialization and pre-training. The experiments are reported in the Appendix B.6 on the updated draft. It can be seen from these experiments that reduced forgetting, by incorporating a larger replay buffer, leads to a better forward transfer. This again validates our main conclusion that less forgetting leads to better forward transfer. We thank the reviewer for suggesting this experiment.
>
> &nbsp;&nbsp;&nbsp;&nbsp;&nbsp;&nbsp; - **Statistical significance of results**: We have updated the plots to show the 95% confidence interval estimates. The results are statistically significant.
>
> 3. **Minor remarks**:
>
> &nbsp;&nbsp;&nbsp;&nbsp;&nbsp;&nbsp;  - **Monotonic ordering of baselines**: Upon reviewer’s suggestion, we have updated the figures to arrange the baselines in the ascending order. It indeed improves the visualization of the figures. Thanks for the suggestion.
>
> &nbsp;&nbsp;&nbsp;&nbsp;&nbsp;&nbsp;  - **Positive forgetting in Figure 3 on CIFAR100**: The reviewer might have misread the Figure 3. The forgetting is positive for CLEAR100, not for CIFAR100. In CLEAR100 all the tasks are very similar -- temporal evolution of the same classes over a period of 10 years. There is a strong positive backward transfer in both CLEAR10 and CLEAR100 where new tasks help learn previous tasks even better owing to the task similarity.
>
> &nbsp;&nbsp;&nbsp;&nbsp;&nbsp;&nbsp;  - **Empirical findings from [Hadsell et al. 2020]**: Since the main point of our paper is to show that forgetting less can be a good inductive bias for forward transfer, we, therefore, used [Hadsell et al. 2020] and [Wolczyk et al. 2021] as canonical references where it was observed that existing focus on catastrophic forgetting by the continual learning community may not lead to models that are better at forward transfer. We did not use any empirical findings from these papers and instead ran all the experiments ourselves.

---

> > ### Comment · Reviewer_gCd6 · 2022-11-17
> > **Final questions to authors**
> >
> > I thank the authors for their thorough response.
> > I'm sorry for my late response here, and I hope the authors would have enough time to answer my few questions below, before I make my final decision.
> >
> > **More important questions:**
> > 1. If I understand correctly, in most plots the only actual CL methods are ER and AGEM (FOMAML and MT are not CL approaches).
> > In that case, it seems like Figure 3 doesn't show any advantage of CL methods (ER and AGEM) over naïve finetuning (FT) in terms of transferability (despite having a better forgetting).
> > Could you please elaborate on what is the community supposed to learn from such results?
> > 1. How did the very loose error bars in Figure 3 become so narrow, comparing to the original version?
> >
> > **Less important questions:**
> > 1. Is there a reason why EWC is only compared-to in the appendix and not in the main plots?
> > 1. How was the $\lambda$ of EWC chosen (in the paper it says $\lambda=100$)?

---

> > > ### Author Response · Authors · 2022-11-17
> > > **Answers to your questions**
> > >
> > > Thanks for the detailed response and continuous engagement with us. Below are the answers to your questions:
> > >
> > > ### More important questions:
> > >
> > > 1. MT and FOMAML are also CL approaches since they will _reuse the model trained on the previous tasks when learning on the current task_, although they have infinite replay buffer size. Note that in practice, we care more about the performance and may not need to restrict the replay buffer size since the storage is cheap. Figure 3 has results on two benchmarks: Split CIFAR-10 and CLEAR100. On Split CIFAR-10, the trend is clear: less forgetting leads to better forward transfer. On CLEAR100, FT, AGEM and ER have roughly the same forward transfer performance while having slightly different forgetting. In this case, our conclusion still holds: forgetting less is a good inductive bias for forward transfer. Our claim is if one method has significantly less forgetting than the other method, then its forward transfer should be significantly better than the other method. If the forgetting is not significantly less, then the forward transfer may not be significantly better. Our findings suggest that in practice, we should use those models that have the smallest forgetting.
> > >
> > > 2. We changed the error bars to be the _95\% confidence interval estimates_ instead of the standard deviation upon the suggestion of reviewer CFvX.
> > >
> > > ### Less important questions:
> > >
> > > 1. EWC is very expensive hence we only tested it on split CIFAR-10 and ResNet18. In the main paper, we used bigger benchmarks and a larger model (ResNet50). That said, we have added a forward reference to EWC results in the main paper.
> > >
> > > 2. For $\lambda$ in EWC, we tried the range $\\{10, 50, 100, 200\\}$ and selected the best one based on the performance on the validation data.
> > >
> > > Let us know if you have any other questions.

---

> > > > ### Comment · Reviewer_gCd6 · 2022-11-18
> > > > **Further discussion**
> > > >
> > > > Perhaps I should make myself clearer regarding my first two questions.
> > > >
> > > > 1. Regarding MT and FOMAML, we can agree to disagree. That's not my main point, but I believe we could agree that *most* research in CL is currently not focused on such cases.
> > > > It is could thus be quite "disappointing" that AGEM&ER offer almost no improvement in transferability comparing to naïve FT (as seen from Figure 3), despite exhibiting much lower forgetting.
> > > > It seems quite crucial for the message of the paper, since it means that in the "limited-memory" CL settings (which again, is more popular), current algorithms *do* improve forgetting but *do not* improve transferability (according to the proposed notion).
> > > >
> > > > 2. I am sorry but I cannot understand your answer regarding the error bars.
> > > > 95% confidence intervals should show ~2 standard deviations (from each side of the mean). Moving from 1 to 2 standard deviations should have made your error bars *worse*, not *better*.
> > > > Could you please clarify how you computed the updated bars?
> > > >
> > > > ----
> > > >
> > > > As a side note, regarding EWC *(just consider it, I'm not expecting further discussion on this)*:
> > > > I understand the computational problems with this method. For the final version of the paper, perhaps you would be able to add experiments with vanilla L2 regularization (where you don't have to approximate the Fisher matrix) to the main manuscript. Two recent papers ("How do Quadratic Regularizers Prevent Catastrophic Forgetting: The Role of Interpolation" and "A Closer Look at Rehearsal-Free Continual Learning") showed that this method's performance is competitive with EWC and MAS.

---

> > > > ### Author Response · Authors · 2022-11-18
> > > > **Further clarification**
> > > >
> > > > We thank the reviewer for the detailed response and continuous engagement with us. Below are some further clarifications:
> > > >
> > > > 1. Figure 3 has results for Split CIFAR-10 and CLEAR100. On Split CIFAR-10, AGEM and ER do have better forward transfer for larger `k` than FT. We assume that the reviewer refers to CLEAR100 where AGEM and ER offer almost no improvement in transferability compared to naïve FT. Note that on CLEAR100, even FT can have positive forgetting, which means learning only on the current task can improve the performance on the previous tasks.  In such a case, FT can have similar forward transfer performance as AGEM and ER. Also, AGEM and ER lead to better forward transfer on other datasets as can be seen in Figures 5, 6, 7.
> > > >
> > > > 2. We should have explained the details in our previous response. 95\% confidence interval is roughly equal to $\bar{x} \pm 1.96 \cdot \text{se}$, where $\bar{x}$ is the sample mean and $\text{se}$ is the standard error (_not standard deviation_). The standard error $\text{se}=\frac{\sigma}{\sqrt{n}}$, where $\sigma$ is the sample standard deviation and $n$ is the sample size. In our experiments, $n=5$. So $1.96 \cdot \text{se} = \frac{1.96}{\sqrt{5}} \cdot \sigma = 0.8765 \cdot \sigma$. Thus, the error bars should be better after the change. We use `seaborn.barplot` (https://seaborn.pydata.org/generated/seaborn.barplot.html) for plotting and set `errorbar=('ci', 95)`.
> > > >
> > > > We hope this clarifies some of the reviewer’s concerns. We can add the L2 regularization baseline in the paper.

---

> ### Comment · Reviewer_gCd6 · 2022-11-18
> **Final decision - Voting for acceptance**
>
> I thank the authors for their responses and fruitful discussion.
>
> ---
>
> I read the other reviews and the author responses and discussed several further issues with the authors.
> Seeing the current version, I believe the paper improved during the reviewing process.
> I believe the paper is interesting and could already benefit the community.
> I think it should be accepted and hence I am raising my score from 5 to 8.
>
> ### Explaining my decision
>
> - Many papers, either theoretical or practical, focus on reducing the forgetting of continual models. Thus, understanding the interplay between this quantity and other quantities of interest, like transferability, is extremely important.
> - I believe the motivation is clearer and that the proposed transferability notion makes more sense now.
> - The authors also fixed several major issues in their experimental methodology and also improved the clarity of the paper.
> - However: It seems like the more "customary" limited-data CL approaches (AGEM, ER, EWC), despite improving the forgetting significantly comparing to fine-tuning (FT), do not improve the transferability (under the proposed notion) *significantly*.
> The authors suggested in their [response](https://openreview.net/forum?id=dL35lx-mTEs&noteId=XuMRm1vEyf) that Figures 5,6,7 *do* show a significant improvement in transferability comparing FT, but I am afraid that in most of these experiments it does not look like the case.
> The paper is still interesting and the correlation is still apparent (especially for non-limited-data approaches like MT and FOMAML), but for **practical** CL algorithms the results might call for a weaker conclusion - that better forgetting *does not* mean worse transferability under the new notion (which is still an interesting conclusion). But it doesn't currently seem like we can conclude that these CL algorithms also imply better transferability.
> (Performing further repetitions can partly solve the significance problem, but it seems like the improvements in transferability are still going to be minor, even when forgetting improves dramatically.)
>
> ---
>
> ### Several final remarks
> Perhaps the authors will find my further comments helpful for the final version (with the additional page, if accepted).
>
> - Like I mentioned before, a regularization method (e.g., EWC) should appear on the figures in the main manuscript as well. If EWC is too costly, then vanilla L2 regularization should also be fine. Also, it should be written somewhere how $\lambda$ was tuned (like the authors explained in our discussion).
> - I believe the ablation study on the replay buffer size (Appendix B.6) should be (at least partly) in the main manuscript (if accepted and there's an additional page).
> - The abstract should also reflect the updates made to the introduction and be clearer regarding the place of this work in the literature.
> - Appendices should be referenced and mentioned in the main manuscript, otherwise they will probably be overlooked.
> - The text in Appendix B.7 is spread across many pages. More generally, some **reordering** is needed in the appendices.
> - In Figure 6, AGEM appears before FT in the middle plot.

---

### Official Review · Reviewer_2ozN · 2022-10-24

**Confidence:** 4
**Correctness:** 3
**Technical Novelty And Significance:** 2
**Empirical Novelty And Significance:** 2
**Recommendation:** 6

**Clarity, Quality, Novelty And Reproducibility:**

The paper is clearly written. Considering that the paper presents an experimental analysis only, the quality of the paper is satisfactory for this type of contribution. The novelty and originality of the paper lies in the research question being evaluated: to what extent forgetting related to forward transfer? The reproducibility of the paper is fair for someone working in this area.

**Strength And Weaknesses:**

Strengths:
- The paper tackles a very important question of whether more (or less) catastrophic forgetting is necessarily associated to more (or less) forward transfer.
- The paper is clearly written, well aligned with literature in the area and easy to follow.

Weaknesses:
- The first main weakness that I find in this paper is in the experiments, in particular the testing of linear probing versus other methods. On page 5, architecture and training details, you mention that baseline models are trained for 50 and 100 epochs when training from scratch, depending on the dataset, and for 20 epochs only when trained from a pre-trained model. When using k-shot linear probing, you train for 100 epochs in all cases. How can you justify that this difference in training iterations is not biasing your results (in this case favouring k-shot linear probing)?
- Although from the experimental results I agree with the statements that "when continual learning experience begins from a randomly initialized model, retaining the knowledge of the past tasks or forgetting less on those tasks is a good inductive bias for forward transfer" and a similar statement for pre-trained models later in the text, I think that these statements are missing a very important aspect: relatedness of tasks. Intuitively, if tasks are not related at all then not forgetting may not influence transfer at all, since only negative transfer will occur to unrelated tasks, or even more forgetting will be beneficial, depending on how much the 'direction' of forgetting is towards the new task; if tasks are related, then not forgetting will be certainly beneficial. In your experimental framework, how is task relatedness considered?

**Summary Of The Paper:**

This paper presents an experimental analysis of catastrophic forgetting versus forward knowledge transfer when tasks are trained continually. The hypothesis posed by the paper is that, contrary to previous findings in the literature, forward transfer does not necessarily imply more forgetting or vice-versa. The authors propose a reformulation of the forward transfer method, which relies on 'linear probing', i.e. training a simple linear classifier on top of a network trained on previous tasks, and testing this simple classifier on the target task test data. The authors also measure feature diversity as a measure of transferability of representations. In the experiments, the authors test their ideas in five benchmark datasets and compare their results to baselines such as finetuning, multitask learning, example-replay methods and metalearning methods. The authors find that less forgetful representations (features) tend to transfer better to new tasks, and these are also more diverse.

**Summary Of The Review:**

The good aspect that I find in this paper is the research question being examined. I think that papers like this, which are focused on examining fundamental/conceptual questions in the area, are certainly required for its progress. On the other hand, I do not find the findings of the paper 100% convincing, and I find the conclusions too broad compared to the extent of the experiments. Although I appreciate the experimental results, I would be convinced if at least one of the following was addressed (or ideally the two of them): 1) more insights into how task relatedness would affect forgetting and transfer (see one of the items I listed as 'weaknesses'); 2) a theoretical analysis supporting the experimental findings. For these reasons, and although I appreciate the effort of the paper on examining important fundamental questions in continual learning, in this first review phase I consider it marginally below the threshold of acceptance.

---

> ### Author Response · Authors · 2022-11-11
> **Response to Reviewer 2ozN**
>
> We thank the reviewer for their feedback. While we address the main concerns / questions of the reviewer below, we would first like to add a meta-comment clarifying the intent of the paper.
>
> One of the main reasons catastrophic forgetting has received so much attention in the continual learning community is the intuition that a model that forgets less on the past tasks accrues more knowledge and should transfer better to new tasks. This intuition is challenged by some recent works. The purpose of our work is to make sure that the continual learning community looks at the tradeoff between forgetting and forward transfer in the right perspective. In this work, we showed that if looked in the right perspective, forgetting less indeed is a good indicator of the forward transfer. We do not claim that previous studies were wrong. We merely say that they looked at the tradeoff in a different perspective which might be useful for end-to-end learning on a fixed benchmark. But in the era of foundation models where models will follow the pretrain-then-finetune paradigm, looking at the forward transfer from continual representation learning perspective, that we are arguing for, is perhaps a better lens. We have made that point more explicit in the updated draft (all the changes are colored in blue). We hope this sets the context for the reviewer to read our rebuttal.
>
> We now address the individual comments:
>
> 1. **How are you making sure that the differences in training iterations are not biasing the results**: K-shot linear probing is a convex optimization problem. So the number of training epochs will not affect the results across baselines as long as we train for a sufficient number of epochs. We found that 100 epochs were more than sufficient for all the baselines to converge for K-shot linear probing.
>
> &nbsp;&nbsp;&nbsp;&nbsp;&nbsp;&nbsp; For the baseline model training, we train for a sufficient number of epochs such that the average learning accuracy across tasks on the validation set doesn't improve as we increase the number of epochs further. For all the baselines, we picked a fixed number of training epochs such that all methods converge. More details are given in the Appendix (A.4).
>
> 2. **Relatedness of tasks**: The reviewer's intuition is correct in that task relatedness will affect the relationship between forgetting and forward transfer. The benchmarks that we considered in this work either have very similar tasks (CLEAR10/ 100), where the same classes are observed over a 10 years period, or unrelated tasks (Split CIFAR10/100, ImageNet), where disjoint classes are observed in each task. In both cases, less forgetting improved the forward transfer, although for more similar tasks the improvement is more significant as intuitively expected. We did not observe that "unrelatedness" of tasks leads to negative transfer. However, if tasks were negatively related to begin with then, yes, less forgetting would intuitively lead to negative transfer. But in our experience negatively related tasks are very rare and in practical machine learning systems many tasks can learn from each other (which is the basis of transfer learning, multitask learning etc.).
>
> &nbsp;&nbsp;&nbsp;&nbsp;&nbsp;&nbsp; That said, we want to emphasize that the point of the paper is precisely to show that when tasks are somewhat related and observed in a continual setting, less forgetting improves forward transfer. It is in this setting that some of the previous works concluded that less forgetting does not improve end-to-end forward transfer making our work non-trivial.

---

> ### Author Response · Authors · 2022-11-17
> **A gentle request for feedback on rebuttal**
>
> Could we kindly ask for the reviewer’s feedback on the rebuttal? We would be happy to incorporate any feedback that the reviewer has.

---

> > ### Comment · Reviewer_2ozN · 2022-11-17
> > **Thanks for the rebuttal**
> >
> > I appreciate the responses of the authors to my concerns.
> >
> > Regarding the first point (number of epochs), I find the experiments more convincing now. Regarding the second point (relatedness of tasks), I would suggest the authors to add this reasoning somewhere in the paper, and perhaps an ablation study on (synthetic) unrelated tasks to show the point - this is just a suggestion.
> >
> > I am quite satisfied with the responses, and after a look at other reviews and responses from the authors, I consider the contribution of the paper reasonably good and therefore I'm increasing my score.

---

> > > ### Author Response · Authors · 2022-11-18
> > > **Added the discussion on task relatedness in the paper**
> > >
> > > We thank the reviewer for responding to our rebuttal. We added the discussion on task relatedness in Appendix (Sec C) (due to space constraints) in the revised draft.

---

### Official Review · Reviewer_pY2z · 2022-10-24

**Confidence:** 2
**Correctness:** 4
**Technical Novelty And Significance:** 2
**Empirical Novelty And Significance:** 3
**Recommendation:** 6

**Clarity, Quality, Novelty And Reproducibility:**

I think the main problem of this paper is the clarity. I think people can reproduce the results with given details.

**Strength And Weaknesses:**

Strength:
- I think the motivation of the paper and the question they try to answer is quite significant for continual learning. Studying the relation between the forgetting and the forward transfer is one of the key aspects of continual learning.

Weaknesses:
- I think the paper lacks motivation that why the proposed definition is better than the existing transferability notions.
- In my opinion, the paper should be written more clearly. The findings do not contradict to the previous findings but it is written in a way that they contradict. I think that is confusing for the readers.

**Summary Of The Paper:**

The paper proposes a new way of measuring forward transfer. They define forward transfer as how easy it is to learn a new task given a set of representations produced by continual learning on previous tasks. And they are looking for an answer to the question of "whether less forgetful representations are more transferable?" And the results show that less forgetful representations lead to a better forward transfer.

**Summary Of The Review:**

The motivation and the research question of the paper is significant and interesting for the community. If the paper is clearly written and explains the differences/similarities with existing approaches more clearly, it could be a much stronger paper.

Update:
After checking other reviews, responses, and the revised version, I change my score to 6. The motivation is more clear now. But I still share the concern of Reviewer 2ozN that the conclusions are too broad compared to the extent of the experiments.

---

> ### Author Response · Authors · 2022-11-11
> **Response to Reviewer pY2z**
>
> We thank the reviewer for their feedback. While we address the main concerns / questions of the reviewer below, we would first like to add a meta-comment clarifying the intent of the paper.
>
> One of the main reasons catastrophic forgetting has received so much attention in the continual learning community is the intuition that a model that forgets less on the past tasks accrues more knowledge and should transfer better to new tasks. This intuition is challenged by some recent works. The purpose of our work is to make sure that the continual learning community looks at the tradeoff between forgetting and forward transfer in the right perspective. In this work, we showed that if looked in the right perspective, forgetting less indeed is a good indicator of the forward transfer. We do not claim that previous studies were wrong. We merely say that they looked at the tradeoff in a different perspective which might be useful for end-to-end learning on a fixed benchmark. But in the era of foundation models where models will follow the pretrain-then-finetune paradigm, looking at the forward transfer from continual representation learning perspective, that we are arguing for, is perhaps a better lens. We have made that point more explicit in the updated draft (all the changes are colored in blue). We hope this sets the context for the reviewer to read our rebuttal.
>
> We now address the individual comments:
>
> **Motivation and writing clarity**: In the revised version, we tried to improve the motivation for the proposed notion of forward transfer. While in the first draft, we tried to emphasize that our study does not contradict the earlier studies (cf. end of the Introduction where we said "... we do not claim that the findings of previous studies (Hadsell et al., 2020; Wolczyk et al., 2021) are in conflict with ours ..."), we have made that point more explicit in the updated draft.
>
> In summary, classical-way of measuring forward transfer via learning accuracy is suitable when learning end-to-end on a fixed benchmark and one knows what future tasks to train on. In the era of foundation models, where pretrain-then-finetune is a more dominant paradigm and one may not know in advance where a foundation model will be finetuned, our measure that looks at the transferability of the representations provides a better notion of forward transfer. Let us know if the clarity is still lacking.

---

> > ### Comment · Reviewer_pY2z · 2022-11-20
> > **Acknowledge**
> >
> > Thank you for your response. I change my score to 6.

---

### Official Review · Reviewer_CFvX · 2022-10-26

**Confidence:** 4
**Correctness:** 3
**Technical Novelty And Significance:** 2
**Empirical Novelty And Significance:** 2
**Recommendation:** 6

**Clarity, Quality, Novelty And Reproducibility:**

# Suggestions


1. I would recommend adding LP baseline that fixes the represetantions throughout learning, and just learns the final classifier for every task i.e. a baseline that removes the second phase fine-tuning of LP-FT. I suspect this baseline would do quite well, especially for the pre-trained models.

2. The authors should report standard error and not standard deviation on the graph.

3.

# Questions:
Why limit LP-FT to the pre-trained setting only?

**Strength And Weaknesses:**

# Strengths
The results in the paper are clear, and support the claims made by the authors---models that forget less demonstrate much better forward transfer performance as per the new definition of forward transfer. Additionally, the paper is clearly written, and all the terms are defined concisely.

The paper includes a reasonable set of baselines, and the empirical studies cover multiple benchmarks. They report results for randomly initialization networks, and pre-trained models.

The work has direct implication for the research in transfer learning, and demonstrate that foundation models that are trained to perform well on the base task trasnfer better.

# Weaknesses

1. If the main contribution of a paper is based on empirical results, it is crucial to fully specify the experimental protocol. The paper doesn't include details on how the hyper-parameters were selected for different methods; did the authors tune the hyper-parameters for each method individually? What were the range of hyper-parameters used for this tuning? Why is the base learning rate different for LP-FT and other methods? The experimental protocol must be fully specified in the paper or the appendix for the paper to be accepted.

2. The primary contribution of the paper is the observation that foundation models that do well on the base task (can do all tasks well with little forgetting) are better at transferring than models that don't perform well on the base task (due to forgetting). The notion of continual learning is not the main emphasis of the paper.

3. The compromise between remembering old information, and forward transfer is a crucial aspect of continual learning in my opinion and should not be ignored by designing an experimenl protocol that hides it. A continual learner system, in almost all cases, cannot divide experience in separate tasks, and learn a seperate predictor for each task.

**Summary Of The Paper:**

The paper demonstrates that mitigating catastrophic forgetting leads to more diverse features that improve the generalization performance on future classification tasks (as measured by learning a classifier on a few data points)

In continual learning literature, learning is often constrained to prevent forgetting on older data; additionally, the model is evaluated on new tasks at the same time it is trying to prevent forgetting on old information; the constraint on future leanring due to the old tasks can degrade the forward transfer performance.

In this paper, the authors propose to isolate the impact remembering old information from the forward transfer performance. They measure the forward trasnfer performance in a separate phase by learning a classifier on top of the fixed representations using a few data point, and evaluate the classifier on a held-out set. After the forward transfer performance is measured, the model is updated on the new task using one of the many continual learning methods.

Finally, the authors show that the models that forget less have more diverse features based on a metric that they propose.

**Summary Of The Review:**


# Closing thoughts
I am personally not convinved that the problem setting that the authors consider, separate tasks with task ids, is the meat of the continual learning problem. I also think that the compromise between remembering past information, and future learning is an important one, and we should design algorithms that can make the right compromise based on experience instead of designing a protocol that ignores the compromise. That said, I still think the paper would be useful for a small audience who do care about task incremental with task id settings.

Once the protocol for hyper-parameter selection is clear and is included, I'm happy to raise my score to an accept because I consider my primary role as to judge the scientific intregity of the work, and not to pass a judgement on the problem setting.

# Update
Changing score from 5 to 6. The authors have added details of hyper-parameter selection, and added a baseline that main claim in the paper more clear and stronger. I think the paper would be a useful addition to the conference.

---

> ### Author Response · Authors · 2022-11-11
> **Response to Reviewer CFvX - Part 1**
>
> We thank the reviewer for their feedback. While we address the main concerns / questions of the reviewer below, we would first like to add a meta-comment clarifying the intent of the paper.
>
> One of the main reasons catastrophic forgetting has received so much attention in the continual learning community is the intuition that a model that forgets less on the past tasks accrues more knowledge and should transfer better to new tasks. This intuition is challenged by some recent works. The purpose of our work is to make sure that the continual learning community looks at the tradeoff between forgetting and forward transfer in the right perspective. In this work, we showed that if looked in the right perspective, forgetting less indeed is a good indicator of the forward transfer. We do not claim that previous studies were wrong. We merely say that they looked at the tradeoff in a different perspective which might be useful for end-to-end learning on a fixed benchmark. But in the era of foundation models where models will follow the pretrain-then-finetune paradigm, looking at the forward transfer from continual representation learning perspective, that we are arguing for, is perhaps a better lens. We have made that point more explicit in the updated draft (all the changes are colored in blue). We hope this sets the context for the reviewer to read our rebuttal.
>
> We now address the individual comments:
>
> 1. **Hyperparameter selection**: We have added a section (see Appendix A.4) in the revised draft on hyper-parameter selection. Below is the summary of how we selected the hyper-parameters for both the continual training and k-shot linear probing.
>
> &nbsp;&nbsp;&nbsp;&nbsp;&nbsp;&nbsp;&nbsp;&nbsp; - **Continual Learning Training**. For different baselines, the shared hyper-parameters are the batch size, the learning rate and the number of training epochs. We do not tune the batch size, but set it to be a fixed number for each benchmark. For the learning rate and the number of training epochs, we choose them based on the average learning accuracy across tasks on the validation data. The range of the learning rate that we consider is $\{0.1, 0.01, 0.001, 0.0001\}$. We found that setting the learning rate to be $0.01$ usually leads to the best average learning accuracy for all the baselines except LP-FT. For LP-FT, we found that setting the learning rate to be $0.001$ leads to better average learning accuracy. We set the number of training epochs to be a sufficiently large number such that the average learning accuracy doesn't improve as we increase the number of epochs further. For all the baselines, we pick a fixed number of training epochs such that all methods converge for each benchmark setting.
>
> &nbsp;&nbsp;&nbsp;&nbsp;&nbsp;&nbsp;&nbsp;&nbsp; - **K-shot Linear Probing Training**. The hyper-parameters are the batch size, the learning rate and the number of training epochs. For each $k$, we just set the batch size to be $\min(k\cdot c, 50)$ and do not tune it. Note that K-shot linear probing is a convex optimization problem. Thus, the number of training epochs will not affect the results across baselines as long as we train for a sufficient number of epochs. We found that $100$ epochs were more than sufficient for all the baselines to converge for K-shot linear probing. Also, the learning rate will not affect the results much as long as we pick a reasonable one. Therefore, we simply fix the learning rate to be $0.01$.
>
>
> 2.  **Continual learning being not the main emphasis of the paper**: Representation learning during continual training is in fact the main emphasis of the paper. Our main claim is that when a **continual learner** builds representations, a less forgetful continual learning algorithm will transfer better to future tasks than a forgetful CL algorithm. Foundation models are only given as an example where this observation can be useful.
>
> 3. **Experimental protocol should not hide the compromise between forgetting and transfer**:  The intention of our work is not to hide the compromise between forgetting and forward transfer. Our protocol highlights the relationship between forgetting and forward transfer when looking at the transferability of representations learned by a continual learner, which could be useful when evaluating different models for transfer learning.
>
> &nbsp;&nbsp;&nbsp;&nbsp;&nbsp;&nbsp; While, in the original draft, we tried to emphasize when the proposed way of looking at forward transfer can be better compared to the classical way that looks at learning accuracy, we have made that point more explicit in the updated version. Please take a look and let us know if we can improve it further.

---

> > ### Comment · Reviewer_CFvX · 2022-11-14
> > **Thank you for the detailed response**
> >
> > Thank you for adding details about hyper-parameter selection; I find the results in the paper more convincing after reading how the experiments were run. I still have some clarifying questions:
> >
> > ### "We found that setting the learning rate to be 0.01 usually leads to the best average learning accuracy for all the baselines except LP-FT."
> >
> > What does 'usually' mean here? I would imagine that if you run a sweep over learning rate, you would know which one is best. It would make sense to do the sweep for every method independently and select the best learning rate for each method. Is that how the experiment was done? I suspect yes, but it would be better to clarify it explicitly in the paper (Usually is too imprecise of a term).
> >
> > ### Addition of the IND baseline
> > I really appreciate the addition of the IND baseline. I think it very clearly shows that the system is doing effective continual learning when starting from random initialization.
> >
> > ### One of the main reasons catastrophic forgetting has received so much attention in the continual learning community is the intuition that a model that forgets less on the past tasks accrues more knowledge and should transfer better to new tasks. This intuition is challenged by some recent works. The purpose of our work is to make sure that the continual learning community looks at the tradeoff between forgetting and forward transfer in the right perspective.
> >
> > I think the intuition is challenged because it is largely incorrect. Whether not forgetting is a good inductive bias or not largely depends on the task. For example, if the targets are non-stationary, such as they are in the RL setting, not forgetting would hurt future performance. On stationary problems, such as supervised learning benchmarks in continual learning, we can expect not forgetting to be a useful inductive bias.
> >
> > My point is that the right perspective for looking at forward transfer vs forgetting is take into account the tradeoff. In any real system, the tradeoff would impact predictions/performance. My intention is not to devalue this work, however. I think this paper asks and answers a clear question scientifically and after the changes, would be a useful addition to the literature. I'm updating my score to reflect that.

---

> > > ### Author Response · Authors · 2022-11-14
> > > **Thanks for the detailed response**
> > >
> > > Thanks for the detailed response. Some clarification:
> > >
> > > Sorry for the confusion. We actually performed hyper-parameter sweeps as the reviewer suggested and observed that setting the learning rate to `0.01` leads to the best average learning accuracy, which was the cross-validation metric, for all the baselines except LP-FT. We have removed the word “usually” and have updated the draft.

---

> ### Author Response · Authors · 2022-11-11
> **Response to Reviewer CFvX - Part 2**
>
> 4. **LP baseline that fixes the representations**: Upon reviewer’s suggestion, we have added the LP baseline, that fixes the representations, in the plots. This baseline was called Independent (IND) in our original draft. It can be seen from the figures that when the continual learning experience begins with a randomly initialized model, independent training performs the worst in terms of forward transfer as it does not accrue any knowledge during continual training. Even when the model is initialized from a pre-trained model, except on split CIFAR-10 benchmark, other baselines that accrue knowledge from the previous tasks do a much better job in terms of forward transfer.
>
> 5. **Error bars**: We have updated the plots to include the 95% confidence interval estimates. Thanks for the suggestion.
>
>
> 6. **Closing thoughts**: We are of the view that in order to understand knowledge accrual during continual learning and how it interacts with the model's ability to learn new tasks quickly, a simplified setup of task-based continual learning was necessary. So that we can probe the model's ability to remember previous tasks and learn new tasks at specified points.
>
> &nbsp;&nbsp;&nbsp;&nbsp;&nbsp;&nbsp; While our intention was not to hide the compromise between forgetting and forward transfer, we have made that point more explicit in the updated draft. We have also added a section for hyper-parameter selection.

---

### Author Response · Authors · 2022-11-10
**Revision summary with rebuttal**

The revised version contains following changes to the original draft (all the changes are color coded in blue).

- **Writing**:

1. [For Reviewer CFvX, pY2z, gCd6] The motivation and contribution of the work have been made more clearer.

2. [For Reviewer CFvX, 2ozN] A new section has been added describing how we select the hyper-parameters.

- **Experiments**: Upon reviewers’ suggestion, following experiments are added,

1. [For Reviewer gCd6] EWC baseline is added.

2. [For Reviewer gCd6] Experience replay baseline with different replay buffer sizes is added to compare the effect of fixing the algorithm and increasing the strength of continual learner to reduce forgetting.

3. [For Reviewer CFvX] LP, where we freeze the representations and just update the classifier, is added (the IND baseline).

We hope these changes will help improve the quality of the manuscript.

---

### Author Response · Authors · 2022-11-19
**Final summary after rebuttal and discussion**

We thank all the reviewers for their detailed reviews and continuous engagement with us. We are aware that such instances are rare and we would like to acknowledge and appreciate that. The discussions with the reviewers helped us to sharpen the message and improve the quality of the paper. We also got an opportunity to clarify some misunderstanding regarding the correctness of our claims.  Below we summarized the main changes made to the first draft during the rebuttal and discussion period (all the changes are color coded in blue).

- **Writing**:

1. [For Reviewer CFvX, pY2z, gCd6] The motivation and contribution of the work have been made more clearer.

2. [For Reviewer CFvX, 2ozN] A new section has been added describing how we select the hyper-parameters.

3. [For Reviewer 2ozN] A discussion on the task relatedness has been added in Appendix C.

- **Experiments**: Upon reviewers’ suggestion, following experiments are added,

1. [For Reviewer gCd6] EWC baseline is added. How the hyper-parameter $\lambda$ in EWC is tuned has been described. See Appendix B.7 for the details.

2. [For Reviewer gCd6] Experience replay baseline with different replay buffer sizes is added to compare the effect of fixing the algorithm and increasing the strength of continual learner to reduce forgetting.

3. [For Reviewer CFvX] LP, where we freeze the representations and just update the classifier, is added (the IND baseline).

---

### Decision · Program_Chairs · 2023-01-20

**Decision:**

Accept: poster

**Justification For Why Not Higher Score:**

The work provided empirical analysis without theoretical support.

**Justification For Why Not Lower Score:**

Overall, this paper receives positive reviews. The reviewers find the technical novelty and contributions are significant enough for acceptance at this conference. The authors' rebuttal helps address the reviewers' concerns and the revised paper is more clear and stronger. The area chair agrees with the reviewers and believe that the work will have positive impacts on transfer learning and continual learning communities.

**Metareview: Summary, Strengths And Weaknesses:**

This paper investigates the relationship between catastrophic forgetting and forward transfer by empirical studies. The authors show that the models forgetting less demonstrate much better forward transfer performance as per the new definition of forward transfer.

Strengths:
1. The research question addressed in the paper is important and interesting.
2. The claim that  the models forgetting less demonstrate much better forward transfer performance are well supported by the experimental results.
3. The work will have positive impact on transfer learning and continual learning
communities.

Weakness:
The work provided empirical analysis without theoretical support.

**Note From Pc:**

if the above contains the word "oral" or "spotlight" please see: "oral" presentation means -> notable-top-5% and "spotlight" means -> notable-top-25%. As stated in our emails, we are disassociating presentation type from AC recommendations